# Revisiting the Role of Pretrained Weights in Model Merging: On Near-Optimality within the Core Subspace

Wenju Sun [1]   Qingyong Li [1]   Tiancheng Li [1]   Yangli-ao Geng [1]   Boyang Albert Li [2]

## Abstract

Model merging offers an efficient solution for integrating task-specific knowledge from multiple fine-tuned models. Most existing approaches focus on manipulating the difference vectors between fine-tuned and pretrained weights, often overlooking the generalization capabilities inherent in the pretrained parameters. In this work, we revisit the role of pretrained weights in model merging and investigate their efficacy from a subspace perspective. We find that the components of pretrained weights residing in the core subspace—defined by the dominant singular vectors—are essential for maintaining generalization across diverse tasks. Specifically, we present empirical evidence that pretrained weights are nearly first-order stationary and exhibit predominantly non-negative curvature within this core subspace with respect to multi-task loss landscapes, indicating near-optimality. These findings suggest that task-specific adaptations should be injected primarily into the orthogonal complement of the core subspace, thereby preserving the generalization properties of the pretrained model. Extensive experiments on vision and vision-language tasks show that this subspace-aware strategy consistently yields improvements over state-of-the-art training-free merging methods, including Task Arithmetic, LOT Merging, ISO, and TSV. The source code is available at https://github.com/SunWenJu123/model-merging.

[1]Key Laboratory of Big Data & Artificial Intelligence in Transportation (Ministry of Education), School of Computer Science and Technology, Beijing Jiaotong University, Beijing, China [2]College of Computing and Data Science, Nanyang Technological University, Singapore. Correspondence to: Yangli-ao Geng <gengyla@bjtu.edu.cn>, Qingyong Li <liqy@bjtu.edu.cn>.

*Proceedings of the 43$^{rd}$ International Conference on Machine Learning*, Seoul, South Korea. PMLR 306, 2026. Copyright 2026 by the author(s).

## 1. Introduction

Large pretrained models have become the cornerstone of modern machine learning, demonstrating remarkable generalization across a spectrum of downstream tasks through fine-tuning (Yang et al., 2025; Zhou et al., 2024). As these models are adapted to increasingly diverse domains, there is a growing need to consolidate task-specific expertise from multiple fine-tuned variants into a single, unified model. Model merging (Li et al., 2023) has emerged as an efficient solution to this challenge, eliminating the need for costly retraining or data aggregation typically required by traditional multi-task learning methods.

Most existing model merging methods are built upon task vectors, represented as the differential vector between parameters of the fine-tuned models and their pretrained origins. Pioneering works such as Task Arithmetic (Ilharco et al., 2023) demonstrate that these vectors encapsulate task-specific knowledge, enabling merged models by linearly combining task vectors and adding the result to the pretrained baseline. Subsequent research has introduced more sophisticated merging techniques to mitigate conflicts among task vectors, including learning adaptive merging weights (Yang et al., 2024b), applying masking or pruning (Yadav et al., 2023; Du et al., 2024; Sun et al., 2025b), and leveraging low-rank or spectral structures to preserve principal task components while suppressing interference (Gargiulo et al., 2025; Marczak et al., 2025; Sun et al., 2025a;c).

Nevertheless, the role of pretrained weights in supporting broad generalization across merged tasks remains underexplored in the literature. With this paper, we ask a fundamental question: To what extent are pretrained parameters already near-optimal for downstream tasks, and how should their generalization be preserved during model merging?

We investigate the role in model merging played by pretrained weights, with a focus on the linear subspace defined by their dominant singular vectors, which we term the *core subspace*. Through the lens of constrained optimization, we provide an analysis of the conditions under which the pretrained weights are close to a local minimum of downstream tasks in the core subspace. Our empirical observations, span-

ning a variety of architectures and tasks, support this claim. Specifically, we demonstrate (1) gradients of downstream objectives are predominantly orthogonal to the core subspace, and (2) the pretrained weights exhibit predominantly non-negative curvature in the core subspace. These two conditions in combination indicate near-optimality with respect to the multi-task loss in the core subspace.

Guided by these findings, we introduce a simple yet effective merging strategy—Core Subspace Preservation (CSP)—which preserves the pretrained model's generalization by projecting merged task vectors onto the orthogonal complement of the core subspace. This training-free approach can be seamlessly incorporated into existing merging frameworks, such as Task Arithmetic (Ilharco et al., 2023), LOT Merging (Sun et al., 2025c), ISO (Marczak et al., 2025), and TSV (Gargiulo et al., 2025). Extensive experiments on vision and vision-language benchmarks demonstrate that CSP consistently improves performance over state-of-the-art training-free merging methods with insignificant additional computational overhead. The contributions of this paper are summarized as follows:

- We analyze the generalization properties of pretrained parameters through a constrained optimization lens and provide empirical evidence that they are near-optimal within the core subspace for model merging.

- We introduce Core Subspace Preservation, a simple yet effective plug-and-play module that preserves pretrained knowledge by projecting merged task vectors onto the orthogonal complement of the core subspace, thereby maintaining generalization during merging.

- We conduct experiments on both vision and vision-language models, demonstrating that Core Subspace Preservation consistently enhances the performance of existing model merging methods with minimal computational overhead.

## 2. Related Works

Model merging aims to integrate multiple task-specific expert models into a single unified model without redundant retraining (Li et al., 2023), which is particularly useful for realistic scenarios such as federated learning (Qi et al., 2025a) or incremental learning (Qi et al., 2025b; Sun et al., 2023). One of the pioneering works is Task Arithmetic (Ilharco et al., 2023), which defines task vectors as the parameter differences between fine-tuned experts and a shared pretrained model, and merges task vectors via linear combination in parameter space. While simple and effective, Task Arithmetic often suffers from performance degradation due to conflicts among task vectors.

To mitigate such interference, numerous methods have been

proposed in recent years. TIES-Merging (Yadav et al., 2023) and PCB Merging (Du et al., 2024) reduce sign conflicts across task vectors, while methods such as CAT Merging (Sun et al., 2025a), TATR (Sun et al., 2025b), and Consensus Merging (Wang et al., 2024) remove outlier or harmful parameters. Another line of work explores test-time adaptation, where merging weights or model components are adjusted using unlabeled test data. Representative methods include AdaMerging (Yang et al., 2024b), Surgery (Yang et al., 2024a), ProDistill (Xu et al., 2025), and mixture-of-experts–based approaches such as WEMoE (Tang et al., 2024) and Twin Merging (Lu et al., 2024). Recent studies increasingly emphasize the principal components of task vectors, such as TSV (Gargiulo et al., 2025) and ISO Merging (Marczak et al., 2025), which improve robustness by discarding less-important components. RegMean (Jin et al., 2023) and LOT Merging (Sun et al., 2025c) leverage representative exemplars to measure the importance of knowledge.

Despite these advances, most existing approaches implicitly treat the pretrained model as a static reference and concentrate on resolving conflicts among task updates. A closely related work (Tang et al., 2026) takes a step in this direction by using the pretrained parameters as the shared MoE trunk, motivated by the observation that dominant principal components of pretrained weights capture substantial downstream-transferable knowledge. In contrast, our primary contribution is a principled explanation of why this design is effective: we formalize a core subspace induced by the pretrained principal directions and show that the pretrained weight is approximately optimal within this core subspace, suggesting that task-specific information should be injected in the orthogonal complement. We also show that this design can be applied on top of existing merging methods to constrain harmful drift in the core subspace, yielding consistent improvements across various datasets and architectures. Another related work (Porrello et al., 2025) assumes the near-optimality of pretrained weights in the context of continual learning. In contrast, our paper investigates this property more explicitly by providing empirical evidence from both first- and second-order conditions to support this claim.

## 3. Preliminary

### 3.1. Problem Setup

This paper considers model merging under the pretraining-finetuning paradigm (Zhou et al., 2024). Let $W_{pre}$ denote the parameters of a model pretrained on a large-scale, diverse corpus. Given a set of $K$ downstream tasks $\hat{\mathcal{T}} = \{T_k \in \mathcal{T}\}_{k=1}^{K}$ where $\mathcal{T}$ represents the task distribution, each task-specific model $W_k$ is obtained by fine-tuning $W_{pre}$ on task $T_k$. Model merging aims to integrate these

task-specific models into a single unified model $W_{mtl}$ that performs well across all tasks. Importantly, the merging process should avoid redundant retraining on the union of datasets to avoid the trivial solution.

### 3.2. Task Vector

A widely adopted framework for model merging relies on task vectors (Ilharco et al., 2023). For each task $T_k$, the task vector is defined as the parameter difference between the fine-tuned model and the pretrained model:

$$\tau_k = W_k - W_{pre}. \tag{1}$$

Note that the arithmetic operations on task vectors and model parameters are performed layer-wise. For simplicity, we analyze each layer independently throughout this paper and treat $\tau_k$ (or $W_k$) as the weight matrix of a given layer.

Task vectors are generally considered to encapsulate the task-specific knowledge. Therefore, existing model merging methods operate on task vectors to compute the merged task vector $\tau_{merge}$. For example, a simple merging strategy, known as Task Arithmetic (Ilharco et al., 2023), sums task vectors linearly:

$$\tau_{merge} = \sum_k \tau_k. \tag{2}$$

The final merged model is then obtained by adding $\tau_{merge}$ with a scalar coefficient $\lambda \in \mathbb{R}$ to the pretrained parameters:

$$W_{merge} = W_{pre} + \lambda \tau_{merge}. \tag{3}$$

## 4. Analysis on Near-Optimality of Pretrained Weights in Core Subspace

Existing work on learning dynamics reveals asymmetric behaviors with respect to different singular values. During training, the directions corresponding to larger singular values of the parameter matrix tend to be learned earlier (Saxe et al., 2013), and the leading principal components of the parameters exhibit stronger generalization across tasks (Lampinen & Ganguli, 2019). In addition, task-specific fine-tuning of pretrained parameters are often concentrated in the subspace associated with small singular values of $W_{pre}$ (Tang et al., 2026).

These observations naturally lead to the following question:

> *Is $W_{pre}$ nearly optimal for $\hat{\mathcal{T}}$ within its large-singular-value subspace?*

In this section, we argue for a positive answer to this question. To facilitate our analysis, we first formally define the core subspace of $W_{pre}$ as the large-singular-value subspace as follows.

**Definition 4.1** (Core Subspace). For the pretrained weight matrix $W_{pre} \in \mathbb{R}^{m \times n}$ with SVD

$$W_{pre} = U\Sigma V^\top, \quad U_r = U[:, 1:r], \ V_r = V[:, 1:r], \tag{4}$$

where $U$ and $V$ are orthogonal matrices, and $\Sigma$ is a diagonal matrix with non-negative singular values $\sigma_1 \geq \sigma_2 \geq \cdots \geq 0$. $r$ is chosen by a fixed rank, which induces "core" directions dominated by large singular values. The core subspace of $W_{pre}$ is defined as

$$\Pi_r(W_{pre}) \triangleq \{W \in \mathbb{R}^{m \times n} | \\ (I - U_r U_r^\top)\, W\, (I - V_r V_r^\top) = 0\}. \tag{5}$$

In other words, for any matrix $W$, $U_r$ and $V_r$ induce a four-block decomposition:

$$W = U_r U_r^\top W V_r V_r^\top + (I - U_r U_r^\top) W V_r V_r^\top \\ + U_r U_r^\top W (I - V_r V_r^\top) + (I - U_r U_r^\top) W (I - V_r V_r^\top). \tag{6}$$

Eq. (5) enforces the last block in the above decomposition to be zero. Equivalently, matrices in $\Pi_r(W_{pre})$ are allowed to vary only along directions associated with the principal singular structure of $W_{pre}$.

We now formalize our hypothesis regarding the optimality of $W_{pre}$ within its core subspace.

**Hypothesis 4.1** (Near-Optimality of Pretrained Weights within the Core Subspace). *Let $\hat{\mathcal{T}}$ denote a finite set of downstream tasks, and define the aggregated objective $L_{mtl}(W) \triangleq \mathbb{E}_{T \sim \hat{\mathcal{T}}}[L_T(W)]$. Assume that the pretrained model has been sufficiently trained on relevant datasets, achieving acceptable but suboptimal performance on $\hat{\mathcal{T}}$. Then, we hypothesize that $W_{pre}$ is near a (local) minimizer of $L_{mtl}(\cdot)$ when the search is constrained to its core subspace $\Pi_r(W_{pre})$, i.e.,*

$$W_{pre} \approx \arg\min_W \ L_{mtl}(W) \quad s.t. \quad W - W_{pre} \in \Pi_r(W_{pre}). \tag{7}$$

### 4.1. Analytical Conditions for Optimality

**The first-order conditions.** We now derive the first-order necessary condition for $W_{pre}$ to be a local minimizer of (7). Consider the Lagrangian

$$\mathcal{L}(W, \Gamma) = L_{mtl}(W) + \langle \Gamma, \ P_U^\perp(W - W_{pre})P_V^\perp \rangle, \tag{8}$$

where $P_U^\perp \triangleq I - U_r U_r^\top$ and $P_V^\perp \triangleq I - V_r V_r^\top$, and $\langle A, B \rangle \triangleq \mathrm{tr}(A^\top B)$ denotes the Frobenius inner product. Here, $\Gamma \in \mathbb{R}^{m \times n}$ is the Lagrange multiplier.

The Karush-Kuhn-Tucker (KKT) first-order necessary conditions for $W_{pre}$ to be a constrained local minimizer of (7) include

$$\nabla_W(L_{mtl}(W) + \langle \Gamma^\star, \ P_U^\perp(W - W_{pre})P_V^\perp \rangle) = 0, \tag{9}$$

$$P_U^\perp(W - W_{pre})P_V^\perp = 0. \tag{10}$$

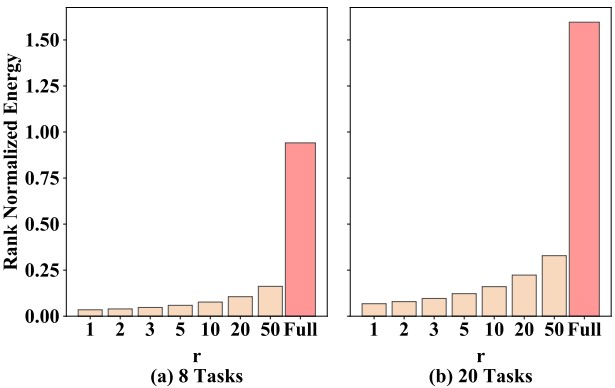

*Figure 1.* Rank-normalized energy of the gradient component projected onto $\Pi_r(W_{pre})$ with different $r$, and the full gradient norm (Full). Experiments are conducted on ViT-B/32 across 8 (a) and 20 (b) downstream tasks.

Substituting $W = W_{pre}$, we see that (10) is trivially true. Further, since $P_U^\perp$ and $P_V^\perp$ are symmetric idempotent projections, the gradient of the constraint inner product with respect to $W$ is

$$\nabla_W \langle \Gamma^\star, P_U^\perp(W - W_{pre})P_V^\perp \rangle = P_U^\perp \Gamma^\star P_V^\perp. \quad (11)$$

Note that $P_U^\perp \Gamma^\star P_V^\perp$ resides in the orthogonal complement of the core subspace $\Pi_r^\perp(W_{pre})$. Therefore, we conclude that $W_{pre}$ is a first-order stationary point if the gradient $\nabla L(W_{pre})$ lies in the orthogonal complement of the core subspace.

**The second-order condition.** As the constraint in Eq. (5) is linear, the second-order condition is entirely determined by the Hessian of the objective function $L_{mtl}$. Therefore, once the first-order stationarity holds, the second-order necessary condition for $W_{pre}$ is that the curvature along all feasible directions should be non-negative:

$$\langle \Delta W, \nabla^2 L_{mtl}(W_{pre})[\Delta W] \rangle \geq 0, \quad \forall \Delta W \in \Pi_r(W_{pre}), \quad (12)$$

where $\nabla^2 L_{mtl}(W_{pre})[\Delta W]$ indicates the Hessian-vector product (HVP) (Pearlmutter, 1994) along the direction $\Delta W$.

Taken together, we reach the following necessary and sufficient condition statement.

> $W_{pre}$ *is a local optimum of the set of downstream tasks in the core subspace if and only if (1) the gradient at $W_{pre}$ has zero projection onto the core subspace; and (2) the Hessian is positive semidefinite on the core space.*

### 4.2. Empirical Evidence

We now provide empirical evidence supporting Hypothesis 4.1.

**Evidence for the first-order condition.** We first verify the first-order necessary condition, i.e., whether the gradient of the multi-task objective at the pretrained weight indeed concentrates outside the core subspace. A direct way is to compare the magnitude of the gradient projected onto the core subspace with that of the full gradient. However, such a comparison is inherently biased: both the effective support of the gradient and the core subspace typically occupy only a small subspace of the entire parameter space, especially when $r$ is small. As a result, raw Frobenius norms are not directly comparable across different subspace dimensions.

To enable a reasonable analysis, we introduce the rank-normalized energy of gradients, defined as

$$\mathcal{E}(G) \triangleq \frac{\|G\|_F^2}{\text{Rank}(G)}, \quad (13)$$

where $G$ denotes a gradient matrix or its projection. This metric measures the average squared gradient magnitude per effective rank, capturing the intrinsic strength of the gradient within its active subspace rather than being dominated by dimensionality effects. In practice, the rank of $G$ is computed using a robust thresholding rule: singular values smaller than $1\%$ of the Frobenius norm are discarded to prevent numerical noise from inflating the rank estimate.

Figure 1 reports the rank-normalized energy of the gradient component projected onto the core subspace $\Pi_r(W_{pre})$ for various values of $r$, along with the corresponding value for the full gradient. We observe that when $r$ is small, the energy within the core subspace remains close to zero compared to that of the full gradient. As $r$ increases, this energy gradually rises, indicating that more gradient is captured as the core subspace expands. In other words, $\nabla_W L_{mtl}(W_{pre})$ is dominated by directions in $\Pi_r^\perp(W_{pre})$, with negligible components in the core subspace. This supports the first-order necessary condition derived in Section 4.1. Additional evidence is reported in Figures 4 and 5.

**Evidence for the second-order condition.** We next investigate the local curvature around $W_{pre}$ in the core subspace. To assess this, we randomly sample perturbation directions $\Delta W \in \Pi_r(W_{pre})$, and compute the HVP $\nabla^2 L_{mtl}(W_{pre})[\Delta W]$ using automatic differentiation. The directional curvature is then quantified by the inner product between the HVP and $\Delta W$:

$$c(\Delta W) \triangleq \langle \Delta W, \nabla^2 L_{mtl}(W_{pre})[\Delta W] \rangle. \quad (14)$$

Figure 2 (a) shows the empirical distribution of $c(\Delta W)$, where approximately $89\%$ of the sampled directions yield non-negative curvature. This observation supports the conclusion that the second-order condition is approximately satisfied within $\Pi_r(W_{pre})$.

**Evidence in the loss landscape.** We now validate the above analysis by examining the loss landscape. Figures 2 (b) and

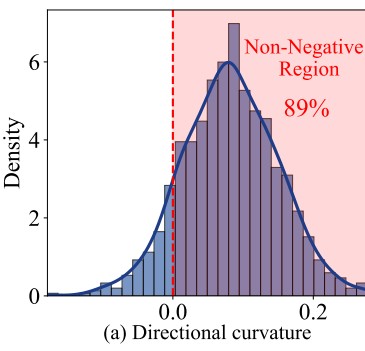 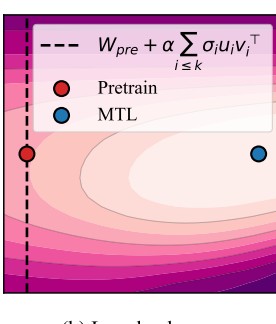 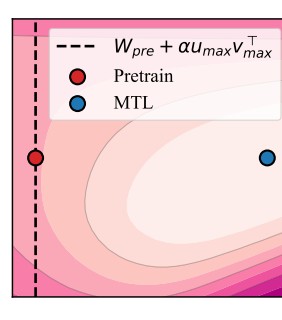 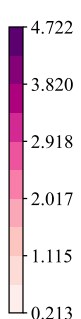

(a) Directional curvature

(b) Loss landscape
(top-k SVD directions)

(c) Loss landscape
(max singular direction)

*Figure 2.* (a) Distribution of curvature evaluated along randomly sampled directions (1,000 random directions) in the core subspace, with the fitted density curve (in blue) showing a Gaussian-like distribution. (b) Multi-task loss landscape spanned by $W_{pre}$, $W_{mtl}$, and $W_{pre} + \alpha \sum_{i \leq r} \sigma_i u_i v_i^\top$. (c) Multi-task loss landscape spanned by $W_{pre}$, $W_{mtl}$, and $W_{pre} + \alpha u_{max} v_{max}^\top$. The dashed lines in (b) and (c) cover the locations traversed as $\alpha$ varies. Experiments are conducted across ViT-B/32 on eight downstream tasks.

(c) visualize the multi-task loss $L_{mtl}$ on a two-dimensional plane spanned by $W_{pre}$ and a multitask-trained reference solution $W_{mtl}$, and an additional probe direction within the core subspace. Specifically, we consider two representative directions, where $\alpha$ is the free variable:

$$W_{(b)}(\alpha) = W_{pre} + \alpha \sum_{i \leq r} \sigma_i u_i v_i^\top$$

$$W_{(c)}(\alpha) = W_{pre} + \alpha u_{\max} v_{\max}^\top, \tag{15}$$

where the first corresponds to a collective perturbation along the top-$r$ singular directions of $W_{pre}$, and the second captures the direction with the largest singular value. For each direction, we plot the loss value $L_{mtl}(W_{(b)}(\alpha))$ and $L_{mtl}(W_{(c)}(\alpha))$ as $\alpha$ varies, which appear as dashed trajectories in subfigures (b) and (c), respectively.

In both cases, the loss along the core-subspace trajectories exhibits a locally convex profile around $W_{pre}$. Specifically, the contour line is nearly tangent to the perturbation line at $W_{pre}$, suggesting that $W_{pre}$ lies close to a local minimum along these directions, with positive and stable curvature. This behavior is not limited to the above two specific probe directions. We traverse the left–right combinations of the top five singular directions within $\Pi_r(W_{pre})$ and consistently observe similar convex loss profiles, indicating that the near-optimality of $W_{pre}$ is a generic property of the core subspace (see Figure 6).

The empirical observations align with our first- and second-order analyses: the gradient component within the core subspace is negligible, and the Hessian is predominantly positive semi-definite on the core subspace.

**Algorithm 1** Core Subspace Preservation

**Input:** pretrained model $W_{pre}$; Merged task vector $\tau^\star$
**Output:** Merged model $W_{mtl}^{csp}$
**for** $l = 1$ **to** $L$ **do**
    // Constructing the core subspace
    $W_{\text{pre}}^{(l)} = U \Sigma V^\top$
    $U_r = U[:, 1:r]$
    $V_r = V[:, 1:r]$
    $P_U^\perp = I - U_r U_r^\top$
    $P_V^\perp = I - V_r V_r^\top$
    // Projecting $\tau^\star$ into the orthogonal complement
    $\Phi^{csp}(\tau^{(l)^\star}) = P_U^\perp \tau^{(l)^\star} P_V^\perp$
// Merging
$\Phi^{csp}(\tau^\star) = \{\Phi^{csp}(\tau^{(1)^\star}), \ldots, \Phi^{csp}(\tau^{(L)^\star})\}$
$W_{mtl}^{csp} = W_{pre} + \lambda \Phi^{csp}(\tau^\star)$
**return** $W_{mtl}^{csp}$

---

The results suggest that $W_{pre}$ *is close to an optimum in the core subspace, as both the first-order condition and the second-order conditions are mostly satisfied, and large-singular-value directions are mostly tangent to the loss contours.*

## 5. Method

In Section 4, we observe a key implication: within the pretrained core subspace, $W_{pre}$ already behaves as a local minimizer of the multi-task objective. In other words, updates along this subspace are unlikely to provide further improvement and may instead disrupt the shared knowledge encoded in the pretrained model. Motivated by this insight, we propose Core Subspace Preservation (CSP), a plug-and-play post-processing module that can be seamlessly integrated into existing model merging methods. CSP

*Table 1.* Multi-task performance when merging ViT-B/32 models on eight vision tasks. Results showing improved performance after integrating the CSP are highlighted in **bold**. The gray values in parentheses denote normalized accuracy (Wang et al., 2024).

| Method | SUN397 | Cars | RESISC45 | EuroSAT | SVHN | GTSRB | MNIST | DTD | Avg Acc |
|---|---|---|---|---|---|---|---|---|---|
| pretrained | 62.3 | 59.7 | 60.7 | 45.5 | 31.4 | 32.6 | 48.5 | 43.8 | 48.0 |
| Individual | 75.3 | 77.7 | 96.1 | 99.7 | 97.5 | 98.7 | 99.7 | 79.4 | 90.5 |
| Traditional MTL | 73.9 | 74.4 | 93.9 | 98.2 | 95.8 | 98.9 | 99.5 | 77.9 | 88.9 |
| Task Arithmetic | 55.2 | 54.9 | 66.7 | 78.9 | 80.2 | 69.7 | 97.3 | 50.4 | 69.1 (75.8) |
| Task Arithmetic w/ ours | **62.9** | **61.5** | **71.8** | **83.4** | 79.5 | **70.5** | 96.8 | **55.1** | **72.7** (80.1) |
| LOT Merging | 67.7 | 67.5 | 85.7 | 94.9 | 93.4 | 89.8 | 98.7 | 63.6 | 82.7 (90.9) |
| LOT Merging w/ ours | 67.7 | 66.5 | **87.0** | **95.5** | **94.6** | **91.6** | **98.9** | **67.0** | **83.6** (91.9) |
| ISO-C | 71.7 | 70.5 | 87.0 | 95.0 | 90.0 | 90.8 | 99.2 | 68.6 | 84.1 (92.7) |
| ISO-C w/ ours | **74.2** | **73.2** | **87.4** | **95.4** | 88.8 | **91.0** | 99.1 | **70.1** | **84.9** (93.8) |
| ISO-CTS | 74.5 | 74.1 | 88.9 | 93.6 | 84.7 | 90.1 | 98.8 | 69.8 | 84.3 (93.2) |
| ISO-CTS w/ ours | 73.7 | 73.8 | 88.7 | **94.7** | **86.2** | **90.8** | **99.0** | 69.6 | **84.7** (93.4) |
| TSV | 70.1 | 72.1 | 85.9 | 94.3 | 90.9 | 91.2 | 99.2 | 68.8 | 84.1 (92.7) |
| TSV w/ ours | **71.1** | **73.4** | **86.3** | **95.4** | **92.3** | **92.6** | **99.3** | **70.7** | **85.1** (93.9) |

*Table 2.* Multi-task performance when merging ViT-L/14 models on eight vision tasks.

| Method | SUN397 | Cars | RESISC45 | EuroSAT | SVHN | GTSRB | MNIST | DTD | Avg Acc |
|---|---|---|---|---|---|---|---|---|---|
| pretrained | 66.8 | 77.7 | 71.0 | 59.9 | 58.4 | 50.5 | 76.3 | 55.3 | 64.5 |
| Individual | 82.3 | 92.4 | 97.4 | 100.0 | 98.1 | 99.2 | 99.7 | 84.1 | 94.2 |
| Traditional MTL | 80.8 | 90.6 | 96.3 | 96.3 | 97.6 | 99.1 | 99.6 | 84.4 | 93.5 |
| Task Arithmetic | 73.9 | 82.1 | 86.6 | 94.1 | 87.9 | 86.7 | 98.9 | 65.6 | 84.5 (89.5) |
| Task Arithmetic w/ ours | **75.1** | **83.6** | **88.0** | **95.6** | **90.2** | **89.9** | **99.1** | **67.8** | **86.2** (91.3) |
| LOT Merging | 76.7 | 88.6 | 91.7 | 98.7 | 97.1 | 95.7 | 99.5 | 76.4 | 90.5 (96.0) |
| LOT Merging w/ ours | 76.3 | **89.1** | **93.1** | **98.9** | **97.2** | **96.5** | **99.6** | **77.6** | **91.0** (96.5) |
| ISO-C | 82.9 | 90.6 | 95.4 | 99.0 | 94.2 | 97.0 | 99.4 | 81.0 | 92.5 (98.2) |
| ISO-C w/ ours | **83.1** | **91.4** | **95.7** | **99.2** | 94.0 | **97.2** | 99.4 | **81.5** | **92.7** (98.5) |
| ISO-CTS | 83.1 | 91.9 | 96.0 | 99.0 | 93.8 | 97.9 | 99.5 | 82.4 | 93.0 (98.8) |
| ISO-CTS w/ ours | **83.3** | 91.8 | **96.1** | **99.5** | **95.1** | **98.5** | 99.5 | **83.0** | **93.4** (99.2) |
| TSV | 79.2 | 89.8 | 93.8 | 98.9 | 95.6 | 96.5 | 99.5 | 80.0 | 91.7 (97.3) |
| TSV w/ ours | **80.5** | **90.2** | **94.2** | **99.4** | **96.5** | **97.8** | **99.7** | **82.1** | **92.5** (98.3) |

preserves the components of pretrained parameters in the core subspace, while only injecting task-specific adaptation in the orthogonal complement.

Given a merged task vector $\tau^\star$, CSP discards the components that lie within the core subspace:

$$\Phi^{csp}(\tau^\star) \triangleq \mathrm{Proj}_{\Pi_r^\perp(W_{pre})}(\tau^\star) = P_U^\perp \tau^\star P_V^\perp. \quad (16)$$

The final merged weight is then given by:

$$W_{mtl}^{csp} = W_{pre} + \lambda \Phi^{csp}(\tau^\star), \quad (17)$$

where $\lambda \in \mathbb{R}$ is the scalar coefficient commonly used in typical model merging methods.

$\tau^\star$ can be obtained from a wide range of task-vector merging strategies, such as Task Arithmetic (Ilharco et al., 2023), TSV (Gargiulo et al., 2025), ISO Merging (Marczak et al., 2025), and LOT Merging (Sun et al., 2025c), among others.

Importantly, CSP offers nearly "free" performance gain when applied on top of them: it requires neither additional data nor any training/fine-tuning procedure, and incurs only a lightweight overhead from performing a layer-wise SVD of $W_{pre}$. The full procedure is outlined in Algorithm 1.

**The selection of hyperparameters ($r$ and $\lambda$).** The hyperparameters $r$ and $\lambda$ constrain $\tau^\star$ from complementary perspectives. Specifically, $\lambda$ limits the magnitude of task vectors, while $r$ restricts their effective dimensionality by preserving the pretrained core subspace. Both hyperparameters are therefore closely tied to the generalization ability of $\tau^\star$: stronger generalization permits a larger $\lambda$ and a smaller $r$. Similarly, scenarios that rely more heavily on generalization, such as larger scaled backbone or a greater number of downstream tasks, typically favor larger values of $r$ (Figures 7 and 8). Moreover, we empirically observe that after applying CSP, the optimal $\lambda$ shifts to larger values

*Table 3.* Average accuracy (%) when merging vision models with more tasks. The checkpoints are obtained from (Tang et al., 2025).

| Backbone | ViT-B/32 | | | ViT-B/16 | | | ViT-L/14 | | |
|---|---|---|---|---|---|---|---|---|---|
| Number of Tasks | 8 | 14 | 20 | 8 | 14 | 20 | 8 | 14 | 20 |
| Task Arithmetic | 69.50 (76.89) | 65.76 (73.63) | 60.63 (68.14) | 77.17 (83.17) | 70.41 (76.81) | 64.46 (70.55) | 82.08 (86.84) | 77.53 (82.72) | 70.98 (75.88) |
| Task Arithmetic w/ ours | **71.50** (78.91) | **67.37** (75.32) | **61.84** (69.36) | **78.28** (84.35) | **71.49** (78.11) | **65.14** (71.21) | **83.60** (88.42) | **78.71** (83.94) | **71.68** (76.61) |
| ISO-C | 80.39 (89.07) | 77.38 (86.41) | 69.94 (77.97) | 85.06 (91.95) | 82.00 (89.41) | 75.71 (82.52) | 90.63 (96.02) | 89.22 (95.34) | 83.18 (88.93) |
| ISO-C w/ ours | **80.38** (89.08) | **78.16** (87.36) | **71.95** (80.28) | **86.71** (93.74) | **82.73** (90.25) | **76.91** (83.79) | **91.28** (96.73) | **89.91** (96.12) | **84.06** (89.88) |
| ISO-CTS | 78.84 (87.49) | 78.12 (87.39) | 73.17 (81.78) | 83.50 (90.32) | 83.61 (91.29) | 77.86 (85.04) | 90.28 (95.67) | 89.60 (95.79) | 84.75 (90.68) |
| ISO-CTS w/ ours | **78.91** (87.58) | **78.41** (87.70) | **73.75** (82.44) | **85.23** (92.78) | **84.45** (92.18) | **78.70** (85.88) | **91.43** (96.90) | **90.36** (96.64) | **84.92** (90.86) |
| TSV | 83.08 (91.63) | 78.38 (87.33) | 73.20 (81.61) | 87.22 (94.15) | 81.77 (89.16) | 77.96 (84.99) | 90.57 (95.88) | 87.84 (93.76) | 82.81 (88.45) |
| TSV w/ ours | **84.43** (93.12) | **79.74** (88.90) | **75.06** (83.67) | **87.93** (95.00) | **82.60** (90.24) | **79.54** (86.62) | **91.64** (97.06) | **88.91** (94.93) | **85.48** (91.28) |

(Figure 3), as conflicting components in task vectors are partially discarded.

# 6. Experimental Results

## 6.1. Settings

The experiments are conducted over vision and vision-language tasks. For vision-only tasks, unless stated otherwise, we perform experiments on eight image classification tasks, using checkpoints from (Ilharco et al., 2023) with ViT-B/32, ViT-B/16, and ViT-L/14 architectures. Additionally, we evaluate performance on 14 and 20 classification tasks, with task vectors sourced from (Tang et al., 2025).

For vision-language tasks, three captioning datasets (COCO Caption (Chen et al., 2015), Flickr30k Caption (Plummer et al., 2015), Textcaps (Sidorov et al., 2020)) and three Visual Question Answering (VQA) datasets (OKVQA (Marino et al., 2019), TextVQA (Singh et al., 2019), and ScienceQA (Lu et al., 2022)) are selected. The BLIP (Shi et al., 2021) model serves as the pretrained model, and we finetune all parameters (including the image encoder, text encoder, and text decoder) to obtain the task vectors.

Since the proposed CSP is a plug-and-play post-processing module, we select Task Arithmetic (Ilharco et al., 2023) and four state-of-the-art methods as the baseline to integrate: LOT Merging (Sun et al., 2025c), ISO-C, ISO-CTS (Marczak et al., 2025), and TSV (Gargiulo et al., 2025). We also report three baseline performances for reference, which are the performance of the pretrained model, the Individual task model, and the Traditional Multi-Task Learning model. Further details are available in the supplementary code.

## 6.2. Comparing Results

**Merging vision-only models.** As shown in Tables 1 and 2, CSP consistently improves the accuracy of several training-free merging algorithms across different backbone scales. On ViT-B/32, CSP significantly enhances weaker baselines (e.g., Task Arithmetic improves from 69.1 to 72.7), while also boosting the performance of state-of-the-art methods.

For example, adding CSP to TSV increases the average accuracy from 84.1 to 85.1, achieving the best overall performance in this setting and enhancing robustness across diverse datasets.

Furthermore, Table 3 studies the scalability of CSP as the number of merged tasks increases. We observe that the performance of all baselines degrades as more tasks are merged, reflecting the increased task interference. Nevertheless, CSP consistently improves all merging methods across 8, 14, and 20 tasks and across all backbone scales. Notably, the relative gains become more pronounced when the number of tasks increases, especially for larger backbones such as ViT-L/14. This suggests that preserving the pretrained core subspace is particularly important for mitigating interference in merging with many tasks.

**Merging vision-language models.** Table 4 reports the results when merging six BLIP models. As can be seen, our CSP improves merging performance on QA tasks. Specifically, adding our method to Task Arithmetic substantially recovers OKVQA and TextVQA accuracy (from 17.71% to 24.67% and from 0.49% to 10.28%), and further strengthens TSV on all three VQA benchmarks, achieving the best results. In contrast, the CSP slightly decreases the performance of the caption tasks. We attribute this issue to the extremely weak zero-shot captioning ability of the pretrained BLIP model. This violates our assumption that the pretrained model provides strong generalization across all downstream tasks. Nevertheless, our method still preserves most downstream knowledge and is particularly effective in mitigating interference on high-conflict VQA tasks.

**Merging language models.** We additionally assess CSP in the context of LLM merging using Qwen3-0.6B (Yang et al., 2025) and Qwen3-8B (Yang et al., 2025). For each model scale, we merge two official fine-tuned variants (Qwen3-Embedding and Qwen3-Reranker) and evaluate performance on two ranking tasks (ArguAna (Wachsmuth et al., 2018), HotpotQA (Yang et al., 2018)) as well as two embedding tasks (BIOSSES (Soğancıoğlu et al., 2017), Banking77 (Casanueva et al., 2020)). Across both model sizes, our

*Table 4.* Multi-task performance when merging BLIP models on six vision-language tasks.

| Method | COCO Caption | Flickr30k Caption | Textcaps | OKVQA | TextVQA | ScienceQA |
|---|---|---|---|---|---|---|
| **Metric** | CIDEr | CIDEr | CIDEr | Accuracy | Accuracy | Accuracy |
| pretrained | 0.07 | 0.03 | 0.05 | 42.80 | 21.08 | 40.50 |
| Individual | 1.17 | 0.65 | 0.65 | 50.84 | 29.79 | 76.89 |
| Task Arithmetic | 0.86 | 0.50 | 0.39 | 17.71 | 0.49 | 40.10 |
| Task Arithmetic w/ ours | 0.83 | 0.46 | 0.38 | **24.67** | **10.28** | **40.18** |
| TSV | 0.84 | 0.55 | 0.38 | 30.02 | 24.38 | 52.65 |
| TSV w/ ours | 0.81 | 0.53 | 0.36 | **35.44** | **25.26** | **55.14** |

*Table 5.* Multi-task performance when merging language models.

| Method | Backbone | ArguAna NDCG@10 | HotpotQA NDCG@10 | BIOSSES Pearson Corr. | Banking77 Accuracy |
|---|---|---|---|---|---|
| Task Arithmetic | Qwen3-0.6B | 0.55 | 0.50 | 0.8324 | 74.67% |
| Task Arithmetic w/ ours | Qwen3-0.6B | **0.57** | **0.56** | **0.8356** | 74.16% |
| TSV | Qwen3-0.6B | 0.61 | 0.60 | 0.8235 | 75.51% |
| TSV w/ ours | Qwen3-0.6B | **0.63** | 0.60 | **0.8266** | **76.03%** |
| Task Arithmetic | Qwen3-8B | 0.63 | 0.64 | 0.8542 | 80.97% |
| Task Arithmetic w/ ours | Qwen3-8B | 0.63 | **0.65** | **0.8625** | **81.43%** |
| TSV | Qwen3-8B | 0.72 | 0.75 | 0.8644 | 84.72% |
| TSV w/ ours | Qwen3-8B | **0.73** | **0.76** | **0.8665** | **84.83%** |

method consistently improves most metrics, indicating that preserving the core subspace effectively mitigates interference and enhances downstream performance during LLM merging. Notably, improvements are observed on both ranking and embedding tasks, demonstrating that CSP remains broadly effective across different scales and task types.

**Merging LoRA-based models.** We further evaluate CSP in a LoRA-based merging setting following (Stoica et al., 2025). The comparison methods include Knots (Stoica et al., 2025) and RobustMerge (Zeng et al., 2025). In this setup, seven ViT-B/32 models are merged (excluding SUN397 due to reproducibility concerns reported in the official repository). As shown in Table 6, our approach improves six out of seven tasks and increases the overall average accuracy from 54.51 to 56.38. The results confirm that CSP can enhance parameter-efficient model merging, helping maintain task-specific knowledge while reducing interference among merged LoRA adapters.

### 6.3. Ablation Study

The core subspace in Definition 4.1 constrains the update directions $\Delta W = W - W_{pre}$ of the merged weight to satisfy $(I - U_r U_r^\top)\Delta W(I - V_r V_r^\top) = 0$, which implicitly couples the left singular subspace $U_r$, the right singular subspace $V_r$, and the corresponding singular values $\Sigma_r$ of the pretrained weight.

To better understand the role of each component, we further decompose the core subspace into three disjoint subspaces:

- **Singular-value subspace**, which preserves both the left and right singular directions, and only allows changes that rescale or reweight the top-$r$ singular values $\Sigma_r$:

$$\{\Delta W | \Delta W = U_r U_r^\top \ \Delta W \ V_r V_r^\top\}. \quad (18)$$

- **Left-singular subspace**, keeps the dominant output directions $V_r$ fixed, while changing the top-$r$ input singular vectors $U_r$:

$$\{\Delta W | \Delta W = (I - U_r U_r^\top) \ \Delta W \ V_r V_r^\top\}. \quad (19)$$

- **Right-singular subspace**, preserves the input directions $U_r$ and introduces perturbations in the output feature space spanned by $V_r$:

$$\{\Delta W | \Delta W = U_r U_r^\top \ \Delta W \ (I - V_r V_r^\top)\}. \quad (20)$$

Table 7 reports an ablation study where we selectively include different subspaces when constructing the core subspace. We observe that removing any single subspace consistently degrades performance across Task Arithmetic and TSV with both ViT-B/32 and ViT-L/14. The best results are obtained when all three subspaces are jointly preserved, validating our core-subspace design. On the other hand, the ablation reveals a clear importance ordering $U_r > V_r > \Sigma_r$. The difference in importance suggests a promising direction for future work that enables fine-grained control of model merging within these subspaces.

*Table 6.* Multi-task performance when merging LoRA-based models on vision tasks.

| Method | Cars | DTD | EuroSAT | GTSRB | MNIST | RESISC45 | SVHN | Average |
|---|---|---|---|---|---|---|---|---|
| KnOTS | 61.31 | 42.87 | 49.44 | 45.13 | 68.47 | 62.65 | 51.66 | 54.51 |
| KnOTS w/ ours | **62.12** | **43.72** | **54.33** | 44.98 | **72.09** | **64.02** | **53.37** | **56.38** |
| RobustMerge | 61.93 | 42.38 | 52.34 | 44.79 | 69.22 | 61.49 | 53.23 | 55.05 |
| RobustMerge w/ ours | **63.01** | 43.44 | **55.53** | 45.19 | **72.34** | 63.39 | **55.87** | **56.97** |

*Table 7.* Ablation study on the impact of different subspaces of the core subspace.

| Singular-value | Left-singular | Right-singular | Task Arithmetic | | TSV | |
|---|---|---|---|---|---|---|
| | | | ViT-B/32 | ViT-L/14 | ViT-B/32 | ViT-L/14 |
| | | | 69.1 | 84.5 | 84.1 | 91.7 |
| | ✓ | ✓ | 72.6 | 85.8 | 85.0 | 92.2 |
| ✓ | | ✓ | 71.1 | 84.7 | 84.2 | 91.6 |
| ✓ | ✓ | | 72.2 | 85.2 | 85.0 | 92.1 |
| ✓ | ✓ | ✓ | **72.7** | **86.2** | **85.1** | **92.5** |

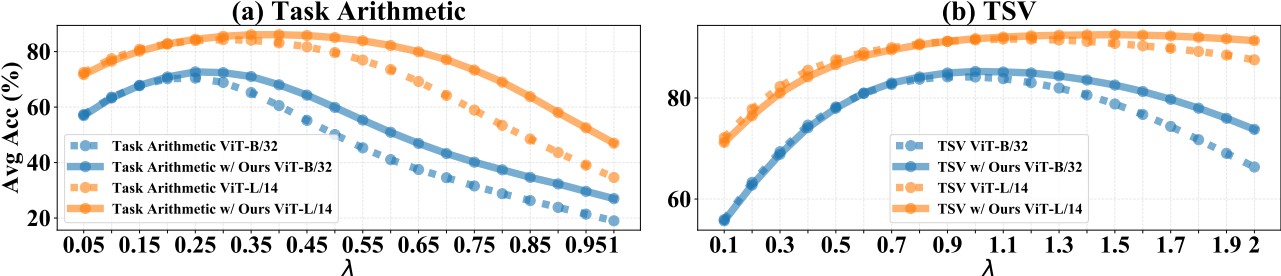

*Figure 3.* Average accuracy of eight vision tasks with different merging coefficient $\lambda$ (cf. Eq. (17)).

### 6.4. Impact on the Merging Coefficient $\lambda$ (cf. Eq. (17))

Figure 3 reports the average accuracy on the eight vision tasks when merging CLIP vision encoders with two backbones (ViT-B/32 and ViT-L/14). We sweep $\lambda$ over a wide range for each method and compare the original baseline against its CSP-augmented variant. Across both Task Arithmetic and TSV, the vanilla baselines exhibit pronounced sensitivity to $\lambda$. When equipped with CSP, the performance curves become flatter and maintain near-optimal accuracy over a broader range of $\lambda$. This indicates that CSP leads the merged model to be less sensitive to the choice of $\lambda$ and thus more robust in practice.

### 7. Conclusion

This paper investigates the generalization of pretrained parameters before model merging, and shows their near-optimality, suggesting they should be preserved during model merging. Our experiments on both vision and vision-language models show that core subspace preservation consistently improves performance with minimal overhead. These results offer insights into the role of the core subspace in pretrained models and suggest a promising direction for

improving robustness in multi-task model merging.

Nevertheless, our method has several limitations. First, its performance depends on selecting an appropriate rank, which may vary across models, tasks, and merging settings. Second, computing SVDs can introduce non-negligible cost at very large scales, especially for models with high-dimensional weight matrices. Third, the benefit of preserving the pretrained core subspace may be reduced when the pretraining distribution and downstream tasks are substantially misaligned. These are possible directions for further research.

### Acknowledgments

This work was supported in part by the Beijing Natural Science Foundation under Grant L231019, by the National Natural Science Foundation of China under Grant 62276019, 62306028, 62501043, by the Fundamental Research Funds for the Central Universities under Grant 2022JBQY007, 2026XKRC005, and by the National Research Foundation Fellowship (NRFF13-2021-0006), Singapore.

## Impact Statement

This paper presents work whose goal is to advance the field of machine learning. There are many potential societal consequences of our work, none of which we feel must be specifically highlighted here.

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

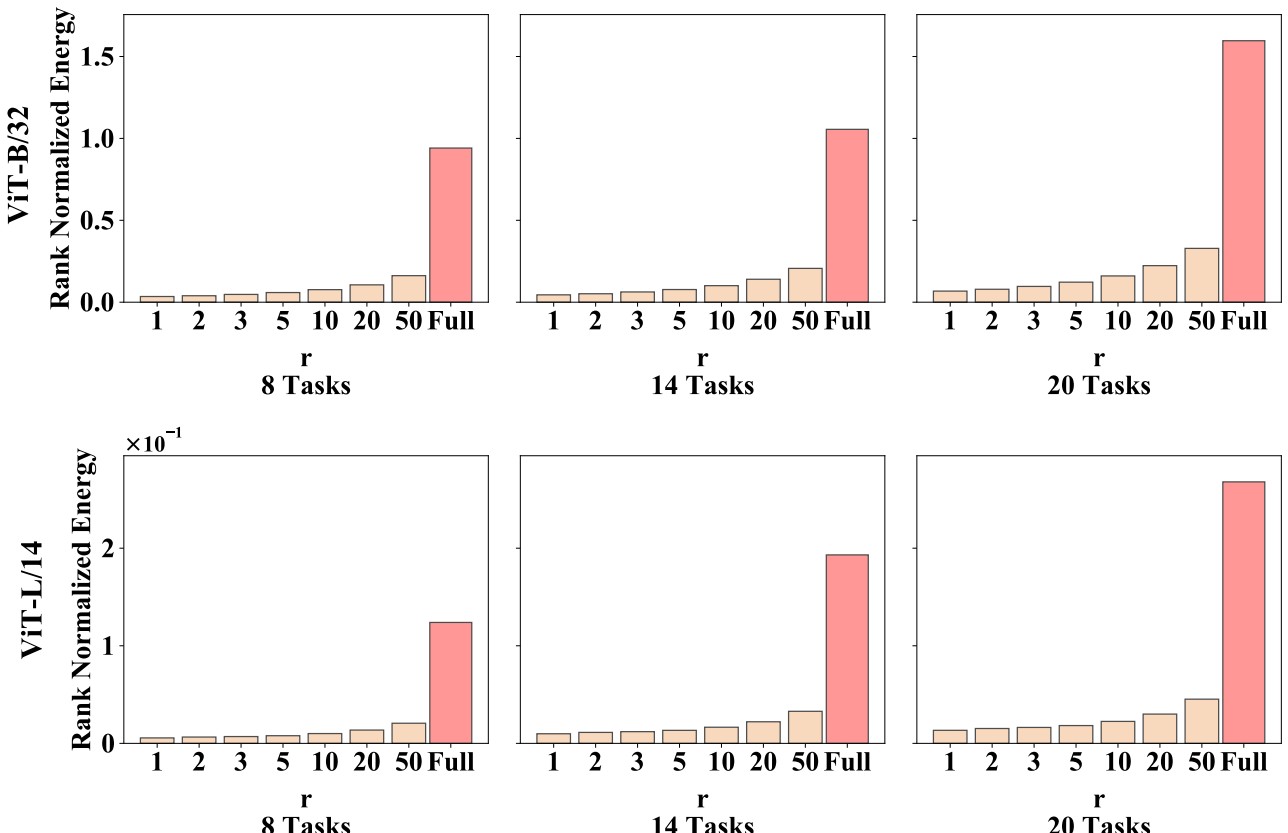

*Figure 4.* Rank-normalized energy of the gradient component projected onto $\Pi_r(W_{pre})$ with different $r$, and the full gradient norm (Full). Experiments are conducted with ViT-B/32 and ViT-L/14 over 8, 14, 20 downstream tasks.

## A. Evidence for the Optimality of the Pretrained Model.

Figures 4, 5, and 6 present various empirical evidence of the near-optimality of the pretrained weights.

Figure 4 shows the rank-normalized energy of the gradient component projected onto $\Pi_r(W_{pre})$ with varying values of $r$. These experiments were conducted using ViT-B/32 and ViT-L/14 on 8, 14, and 20 downstream tasks, demonstrating how the energy distribution of the gradient changes across different model configurations. Figure 5 zooms in on the energy distribution across different types of layers in ViT-B/32. We can see that in the core subspace, the gradient consistently vanishes with small $r$, suggesting that the first-order properties of the model are well established as expected.

Moreover, Figure 6 visualizes the loss landscapes by sweeping perturbation directions from pairwise combinations of the top-5 singular vectors, computed on ViT-B/32 using the multi-task objective over 8 downstream vision tasks. The perturbation lines are observed to be tangent to the contours of the loss surface, indicating that the second-order properties of the pretrained weights also hold for most cases.

## B. Experiment Details

This section summarizes the experimental setup, including compute environment, evaluation benchmarks, and the model-merging baselines used throughout the paper.

### B.1. Environment

All experiments in the main paper and appendix were run on a single Ubuntu 16.04 workstation with 18 Intel Xeon CPUs (2.60 GHz), 256 GB RAM, and 8 NVIDIA RTX 4090 GPUs. All implementations were in Python 3.8.

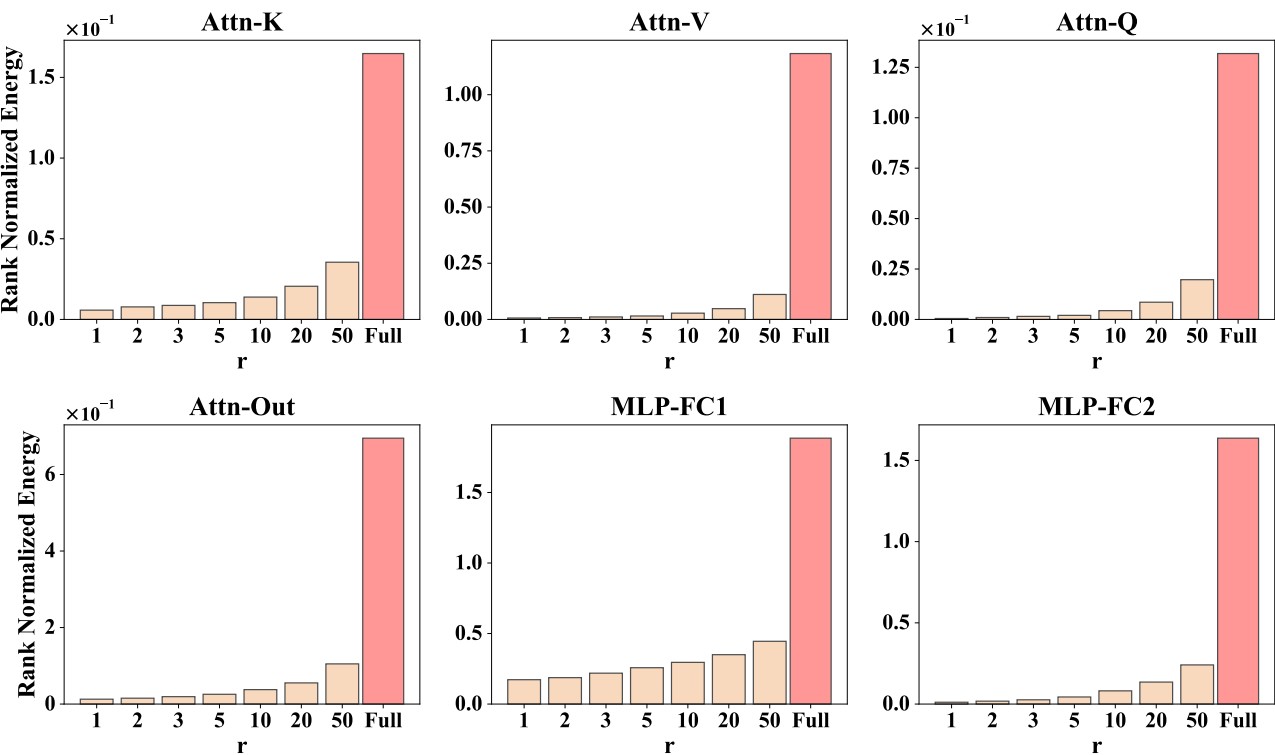

*Figure 5.* Rank-normalized energy of the gradient component projected onto $\Pi_r(W_{pre})$ with different $r$, and the full gradient norm (Full). Experiments are conducted over various types of layers of ViT-B/32 over 8 downstream tasks.

### B.2. Datasets

We evaluate across vision-only, vision-language tasks. This section provides the details of the involved datasets.

**Vision tasks (image classification).** We consider eight standard classification benchmarks to assess the quality of the merged vision model:

- **SUN397** (Xiao et al., 2016): 108,754 images spanning 397 scene categories, with at least 100 images per class.

- **Stanford Cars (Cars)** (Krause et al., 2013): 16,185 images from 196 car classes, with an approximately even train/test split.

- **RESISC45** (Cheng et al., 2017): 31,500 remote-sensing images over 45 scene types (about 700 images per class).

- **EuroSAT** (Helber et al., 2019): 27,000 geo-referenced satellite images labeled into 10 classes.

- **SVHN** (Netzer et al., 2011): 10-way digit recognition from street-view imagery, with 73,257 training and 26,032 test images, plus 531,131 additional training samples.

- **GTSRB** (Stallkamp et al., 2011): 43-category traffic sign recognition with over 50,000 images.

- **MNIST** (LeCun & Cortes, 2010): handwritten digit classification with 60,000 training and 10,000 test images across 10 classes.

- **DTD** (Cimpoi et al., 2014): 5,640 texture images from 47 categories (roughly 120 images per class).

**Vision-language tasks (captioning and VQA).** For captioning, we use three widely adopted datasets; for VQA, we evaluate on three benchmarks emphasizing general reasoning, reading text in the wild, and science reasoning:

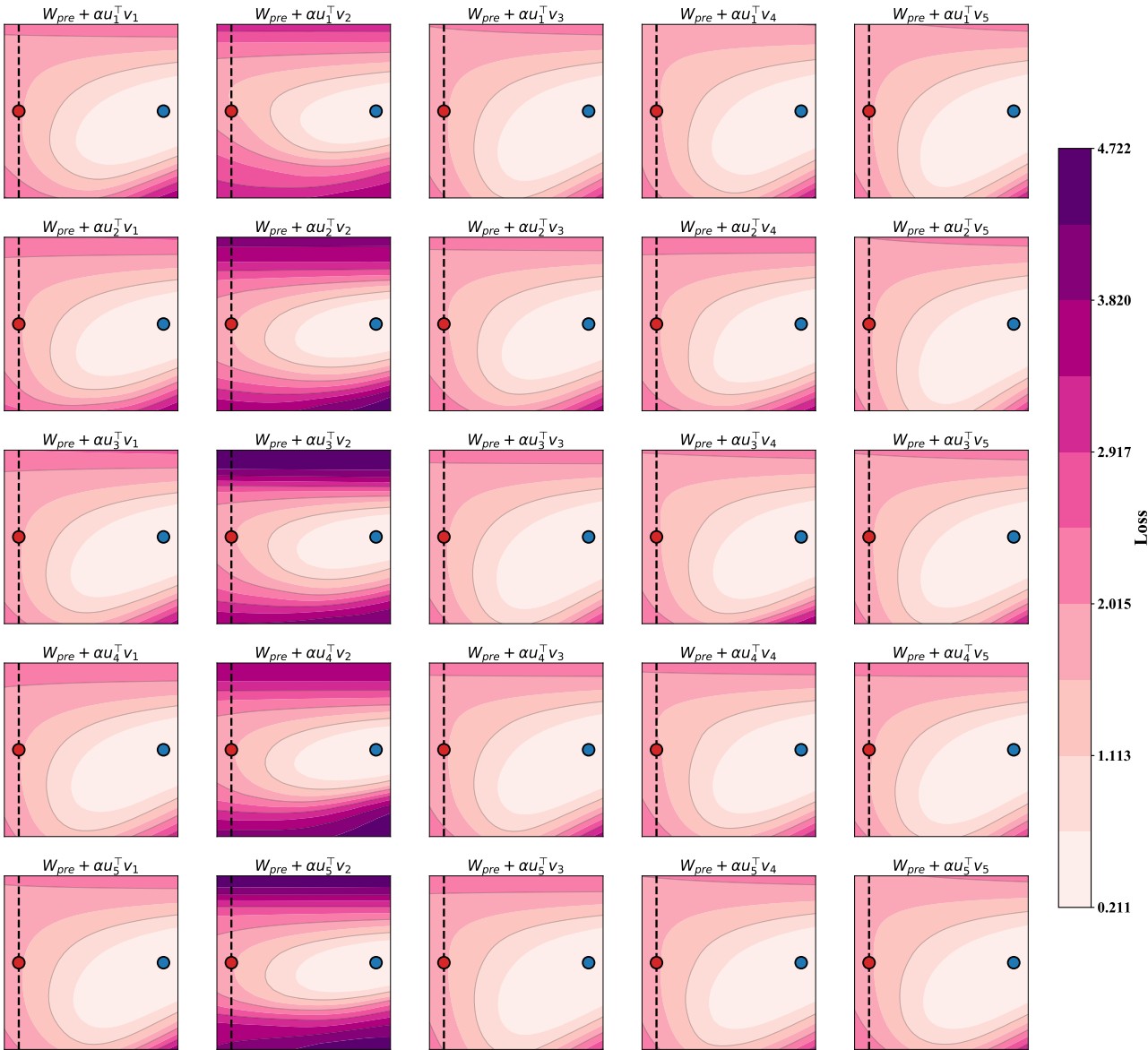

*Figure 6.* Loss landscapes by sweeping perturbation directions from pairwise combinations of the top-5 singular vectors. All landscapes are computed on ViT-B/32 using the multi-task objective over 8 downstream tasks.

*Table 8.* Multi-task performance when merging ViT-B/32 models on eight vision tasks.

| Method | SUN397 | Cars | RESISC45 | EuroSAT | SVHN | GTSRB | MNIST | DTD | Avg Acc |
|---|---|---|---|---|---|---|---|---|---|
| *Basic baseline methods* | | | | | | | | | |
| pretrained | 62.3 | 59.7 | 60.7 | 45.5 | 31.4 | 32.6 | 48.5 | 43.8 | 48.0 |
| Individual | 75.3 | 77.7 | 96.1 | 99.7 | 97.5 | 98.7 | 99.7 | 79.4 | 90.5 |
| Traditional MTL | 73.9 | 74.4 | 93.9 | 98.2 | 95.8 | 98.9 | 99.5 | 77.9 | 88.9 |
| *Test-time training-based methods* | | | | | | | | | |
| AdaMerging | 64.5 | 68.1 | 79.2 | 93.8 | 87.0 | 91.9 | 97.5 | 59.1 | 80.1 |
| AdaMerging++ | 66.6 | 68.3 | 82.2 | 94.2 | 89.6 | 89.0 | 98.3 | 60.6 | 81.1 |
| Surgery Merging | 63.8 | 59.9 | 83.3 | 97.9 | 87.0 | 87.0 | 98.6 | 69.4 | 80.9 |
| Localize-and-Stitch | 67.2 | 68.3 | 81.8 | 89.4 | 87.9 | 86.6 | 94.8 | 62.9 | 79.9 |
| DOGE AM | 70.5 | 74.8 | 88.7 | 94.1 | 91.6 | 95.7 | 98.8 | 72.5 | 85.9 |
| *Training-free methods* | | | | | | | | | |
| Weight Averaging | 65.3 | 63.4 | 71.4 | 71.7 | 64.2 | 52.8 | 87.5 | 50.1 | 65.8 |
| Fisher Merging | 68.6 | 69.2 | 70.7 | 66.4 | 72.9 | 51.1 | 87.9 | 59.9 | 68.3 |
| RegMean | 65.3 | 63.5 | 75.6 | 78.6 | 78.1 | 67.4 | 93.7 | 52.0 | 71.8 |
| Task Arithmetic | 55.2 | 54.9 | 66.7 | 78.9 | 80.2 | 69.7 | 97.3 | 50.4 | 69.1 |
| Task Arithmetic w/ ours | **62.9** | **61.5** | **71.8** | **83.4** | 79.5 | **70.5** | 96.8 | **55.1** | **72.7** |
| Ties-Merging | 59.8 | 58.6 | 70.7 | 79.7 | 86.2 | 72.1 | 98.3 | 54.2 | 72.4 |
| TATR | 62.7 | 59.3 | 72.3 | 82.3 | 80.5 | 72.6 | 97.0 | 55.4 | 72.8 |
| Ties-Merging & TATR | 66.3 | 65.9 | 75.9 | 79.4 | 79.9 | 68.1 | 96.2 | 54.8 | 73.3 |
| Consensus Merging | 65.7 | 63.6 | 76.5 | 77.2 | 81.7 | 70.3 | 97.0 | 57.1 | 73.6 |
| AWD Merging | 63.5 | 61.9 | 72.6 | 84.9 | 85.1 | 79.1 | 98.1 | 56.7 | 75.2 |
| PCB Merging | 63.8 | 62.0 | 77.1 | 80.6 | 87.5 | 78.5 | 98.7 | 58.4 | 75.8 |
| CAT Merging | 68.1 | 65.4 | 80.5 | 89.5 | 85.5 | 78.5 | 98.6 | 60.7 | 78.3 |
| DOGE TA | 67.7 | 70.1 | 82.0 | 90.3 | 86.3 | 86.8 | 98.3 | 64.0 | 80.7 |
| LOT Merging | 67.7 | 67.5 | 85.7 | 94.9 | 93.4 | 89.8 | 98.7 | 63.6 | 82.7 |
| FR-Merging | 66.2 | 64.5 | 77.2 | 90.1 | 85.4 | 82.3 | 98.5 | 60.0 | 78.1 |
| LOT Merging | 67.7 | 67.5 | 85.7 | 94.9 | 93.4 | 89.8 | 98.7 | 63.6 | 82.7 |
| LOT Merging w/ ours | 67.7 | 66.5 | **87.0** | **95.5** | **94.6** | **91.6** | **98.9** | **67.0** | **83.6** |
| ISO-C | 71.7 | 70.5 | 87.0 | 95.0 | 90.0 | 90.8 | 99.2 | 68.6 | 84.1 |
| ISO-C w/ ours | **74.2** | **73.2** | **87.4** | 95.4 | 88.8 | **91.0** | 99.1 | **70.1** | **84.9** |
| ISO-CTS | 74.5 | 74.1 | 88.9 | 93.6 | 84.7 | 90.1 | 98.8 | 69.8 | 84.3 |
| ISO-CTS w/ ours | 73.7 | 73.8 | 88.7 | **94.7** | **86.2** | **90.8** | **99.0** | 69.6 | **84.7** |
| TSV | 70.1 | 72.1 | 85.9 | 94.3 | 90.9 | 91.2 | 99.2 | 68.8 | 84.1 |
| TSV w/ ours | **71.1** | **73.4** | **86.3** | **95.4** | **92.3** | **92.6** | **99.3** | **70.7** | **85.1** |

- **COCO Caption** (Chen et al., 2015): MS COCO captioning corpus with over 330,000 images and five human-written captions per image.

- **Flickr30k Caption** (Plummer et al., 2015): 31,000 images, each paired with five descriptive sentences, commonly used for captioning and image–text retrieval.

- **TextCaps** (Sidorov et al., 2020): an OCR-centric caption benchmark with 145,000 image–caption pairs, requiring recognition of embedded scene text.

- **OKVQA** (Marino et al., 2019): knowledge-intensive VQA with over 14,000 questions that typically require external/world knowledge beyond the image.

- **TextVQA** (Singh et al., 2019): text-focused VQA with over 45,000 questions on 28,000 images, requiring OCR and joint visual-text reasoning.

- **ScienceQA** (Lu et al., 2022): multimodal multiple-choice QA with over 21,000 questions across science domains, paired with images and textual rationales.

*Table 9.* Multi-task performance when merging ViT-L/14 models on eight vision tasks.

| Method | SUN397 | Cars | RESISC45 | EuroSAT | SVHN | GTSRB | MNIST | DTD | Avg Acc |
|---|---|---|---|---|---|---|---|---|---|
| *Basic baseline methods* | | | | | | | | | |
| pretrained | 66.8 | 77.7 | 71.0 | 59.9 | 58.4 | 50.5 | 76.3 | 55.3 | 64.5 |
| Individual | 82.3 | 92.4 | 97.4 | 100.0 | 98.1 | 99.2 | 99.7 | 84.1 | 94.2 |
| Traditional MTL | 80.8 | 90.6 | 96.3 | 96.3 | 97.6 | 99.1 | 99.6 | 84.4 | 93.5 |
| *Test-time training-based methods* | | | | | | | | | |
| AdaMerging | 79.0 | 90.3 | 90.8 | 96.2 | 93.4 | 98.0 | 99.0 | 79.9 | 90.8 |
| AdaMerging++ | 79.4 | 90.3 | 91.6 | 97.4 | 93.4 | 97.5 | 99.0 | 79.2 | 91.0 |
| Surgery Merging | 75.7 | 84.4 | 93.1 | 98.8 | 91.3 | 93.4 | 99.1 | 76.1 | 89.0 |
| Localize-and-Stitch | 74.4 | 78.0 | 86.0 | 94.6 | 93.4 | 92.5 | 98.5 | 74.9 | 86.5 |
| DOGE AM | 79.7 | 91.6 | 94.4 | 96.7 | 96.5 | 98.6 | 99.0 | 84.1 | 92.6 |
| *Training-free methods* | | | | | | | | | |
| Weight Averaging | 72.1 | 81.6 | 82.6 | 91.9 | 78.2 | 70.7 | 97.1 | 62.8 | 79.6 |
| Fisher Merging | 69.2 | 88.6 | 87.5 | 93.5 | 80.6 | 74.8 | 93.3 | 70.0 | 82.2 |
| RegMean | 73.3 | 81.8 | 86.1 | 97.0 | 88.0 | 84.2 | 98.5 | 60.8 | 83.7 |
| Task Arithmetic | 73.9 | 82.1 | 86.6 | 94.1 | 87.9 | 86.7 | 98.9 | 65.6 | 84.5 |
| Task Arithmetic w/ ours | **75.1** | **83.6** | **88.0** | **95.6** | **90.2** | **89.9** | **99.1** | **67.8** | **86.2** |
| Ties-Merging | 76.5 | 85.0 | 89.3 | 95.7 | 90.3 | 83.3 | 99.0 | 68.8 | 86.0 |
| TATR | 74.6 | 83.7 | 87.6 | 93.7 | 88.6 | 88.1 | 99.0 | 66.8 | 85.3 |
| Ties-Merging & TATR | 76.3 | 85.3 | 88.8 | 94.4 | 90.8 | 88.7 | 99.2 | 68.8 | 86.5 |
| Consensus Merging | 75.0 | 84.3 | 89.4 | 95.6 | 88.3 | 82.4 | 98.9 | 68.0 | 85.2 |
| AWD Merging | 76.2 | 85.4 | 88.7 | 96.1 | 92.4 | 92.3 | 99.3 | 69.4 | 87.5 |
| PCB Merging | 76.2 | 86.0 | 89.6 | 95.9 | 89.9 | 92.3 | 99.2 | 71.4 | 87.6 |
| CAT Merging | 78.7 | 88.5 | 91.1 | 96.3 | 91.3 | 95.7 | 99.4 | 75.7 | 89.6 |
| DOGE TA | 76.7 | 87.7 | 91.6 | 96.2 | 94.4 | 93.4 | 98.9 | 71.6 | 88.8 |
| LOT Merging | 76.7 | 88.6 | 91.7 | 98.7 | 97.1 | 95.7 | 99.5 | 76.4 | 90.5 |
| FR-Merging | 76.4 | 87.0 | 90.2 | 96.8 | 92.0 | 92.8 | 99.3 | 71.5 | 88.3 |
| LOT Merging | 76.7 | 88.6 | 91.7 | 98.7 | 97.1 | 95.7 | 99.5 | 76.4 | 90.5 |
| LOT Merging w/ ours | 76.3 | **89.1** | **93.1** | **98.9** | **97.2** | **96.5** | **99.6** | **77.6** | **91.0** |
| ISO-C | 82.9 | 90.6 | 95.4 | 99.0 | 94.2 | 97.0 | 99.4 | 81.0 | 92.5 |
| ISO-C w/ ours | **83.1** | **91.4** | **95.7** | **99.2** | 94.0 | **97.2** | 99.4 | **81.5** | **92.7** |
| ISO-CTS | 83.1 | 91.9 | 96.0 | 99.0 | 93.8 | 97.9 | 99.5 | 82.4 | 93.0 |
| ISO-CTS w/ ours | **83.3** | 91.8 | **96.1** | **99.5** | **95.1** | **98.5** | 99.5 | **83.0** | **93.4** |
| TSV | 79.2 | 89.8 | 93.8 | 98.9 | 95.6 | 96.5 | 99.5 | 80.0 | 91.7 |
| TSV w/ ours | **80.5** | **90.2** | **94.2** | **99.4** | **96.5** | **97.8** | **99.7** | **82.1** | **92.5** |

## B.3. Baselines

We implement and compare against several representative model-merging approaches, integrating each baseline with our experimental pipeline:

- **Task Arithmetic** (Ilharco et al., 2023) represents each task by a task vector (the parameter difference between a task expert and the shared pretrained model), and forms a multi-task model by adding a weighted combination of task vectors to the pretrained parameters.

- **LOT Merging** (Sun et al., 2025c) merges task experts by encouraging feature-level consistency, aligning intermediate representations of the merged model with those of the task-specific experts.

- **ISO-C** (Marczak et al., 2025) constructs a common merging subspace from Task Arithmetic by taking the SVD of the summed task matrix, then flattens the singular-value spectrum (i.e., uses isotropic scaling) to reduce dominance of a few directions and improve cross-task balance.

- **ISO-CTS** (Marczak et al., 2025) extends ISO-C by retaining top common directions while augmenting them with task-specific residual directions (obtained after projecting out the common subspace). The combined basis is orthogonalized

*Table 10.* Computational complexity comparison (in seconds) when merging ViT-B/32 and ViT-L/14 models across eight vision tasks, measured on a single RTX 3090 GPU.

| Method | LOT Merging | ISO-C | ISO-CTS | TSV | CSP |
|--------|-------------|-------|---------|-----|-----|
| ViT-B/32 | 44 | 11 | 41 | 28 | 5 |
| ViT-L/14 | 161 | 29 | 134 | 95 | 10 |

*Table 11.* Comparison of different strategies for selecting core space.

| Base Methods | Core Space | ViT-B/32 | ViT-L/14 |
|--------------|------------|----------|----------|
| ISO-CTS | Top-$r$ | 84.7 | **93.4** |
| ISO-CTS | Top-$p$ | **84.8** | **93.4** |
| TSV | Top-$r$ | 85.1 | **92.5** |
| TSV | Top-$p$ | **85.2** | **92.5** |

and isotropically scaled, improving robustness when merging many diverse tasks.

- **TSV** (Gargiulo et al., 2025) computes a merged task vector aligned with principal components of per-task updates, prioritizing high-variance directions intended to capture informative shared structure.

## C. Additional Experiments

### C.1. Comparison Results

This section reports a comprehensive comparison against recent state-of-the-art model-merging approaches, covering both test-time training methods and training-free methods. For test-time training baselines, we evaluate AdaMerging and AdaMerging++ (Yang et al., 2024b), Surgery (Yang et al., 2024a), Localize-and-Stitch (He et al., 2025), and DOGE AM (Wei et al., 2025). For training-free baselines, we include Weight Averaging, Fisher Merging (Matena & Raffel, 2022), RegMean (Jin et al., 2023), Task Arithmetic (Ilharco et al., 2023), Ties-Merging (Yadav et al., 2023), TATR (Sun et al., 2025b), Consensus Merging (Wang et al., 2024), AWD Merging (Xiong et al., 2024), PCB Merging (Du et al., 2024), CAT Merging (Sun et al., 2025a), DOGE TA (Wei et al., 2025), LOT Merging (Sun et al., 2025c), FR-Merging (Zheng & Wang, 2025), and TSV (Gargiulo et al., 2025). We report results for merging eight task-specific experts based on ViT-B/32 and ViT-L/14 backbones in Tables 8 and 9, respectively.

### C.2. Analysis of Computational Complexity

Table 10 compares the wall-clock time (seconds) required to merge ViT-B/32 and ViT-L/14 on a single RTX 3090. Among the evaluated approaches, ISO-C is the most efficient on both backbones, whereas LOT Merging and ISO-CTS are more expensive. Notably, all merging methods incur markedly higher runtime than CSP (5s for ViT-B/32 and 10s for ViT-L/14), indicating that CSP can yield performance improvements with negligible additional computational overhead.

### C.3. Sensitivity to the hyperparameter $r$

The hyper-parameter $r$ controls the dimensionality of the core subspace, i.e., the number of dominant singular directions of the pretrained weight that are preserved during model merging. Intuitively, a larger $r$ retains more pretrained directions, while a smaller $r$ imposes a stronger constraint by projecting merged task vectors into a lower-dimensional orthogonal complement. Therefore, $r$ trades off between preserving pretrained generalization and allowing task-specific adaptation.

Figures 7 and 8 investigate the sensitivity of our method to different values of $r$. Overall, we observe that the performance consistently improves across a wide range of $r$, indicating that the proposed method is relatively robust to this hyperparameter. However, the optimal choice of $r$ varies across different settings.

From Figure 7, we find that larger architectures generally benefit from larger values of $r$. This suggests that more expressive models contain richer pretrained representations distributed over a higher-dimensional core subspace, and thus require preserving more singular directions to maintain their generalization capability.

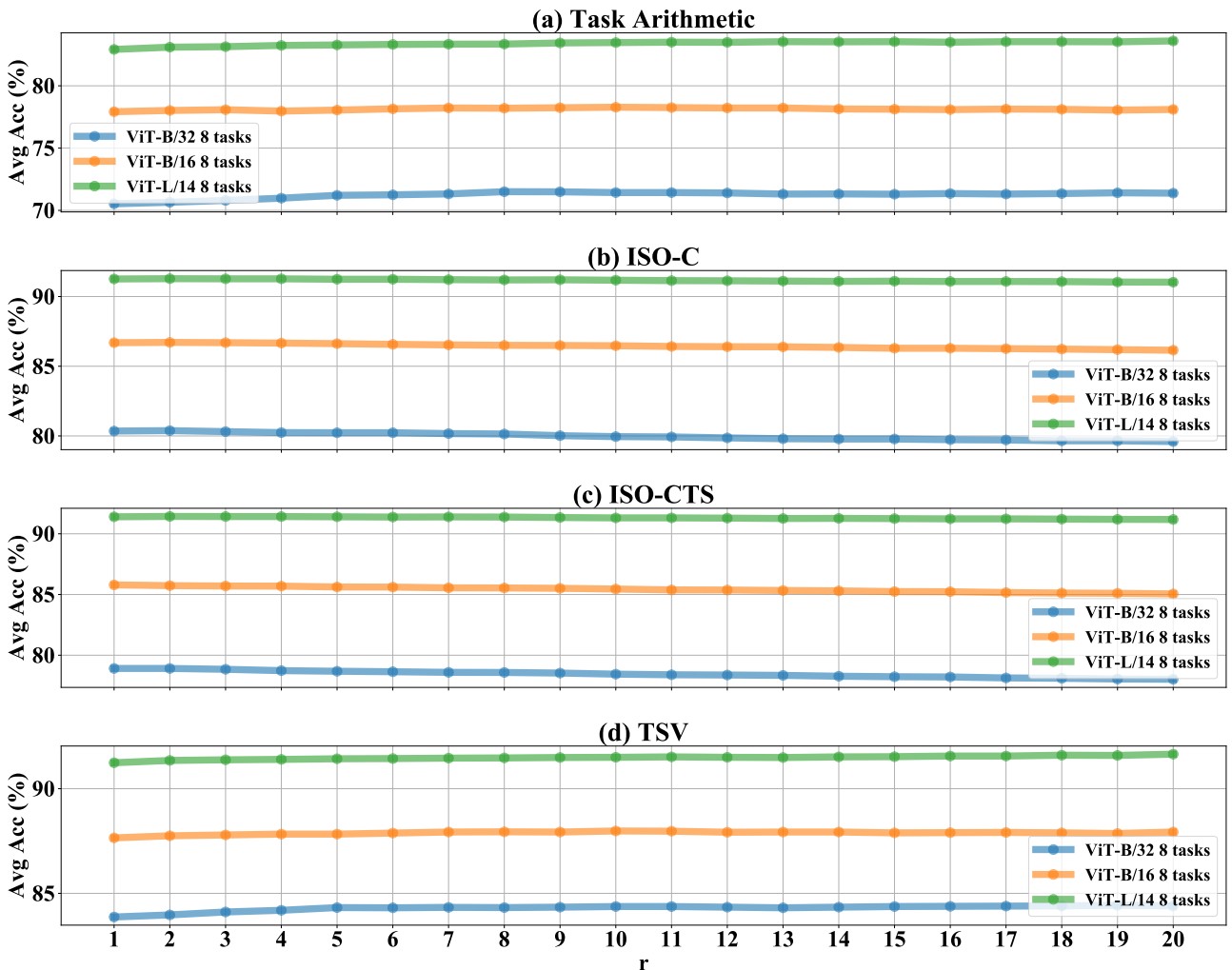

*Figure 7.* Average accuracy across different architectures with various $r$.

Figure 8 further shows that the optimal $r$ tends to increase with the number of tasks. When merging a larger set of tasks, the merged task vectors introduce more diverse and potentially conflicting updates, and preserving a larger core subspace helps stabilize the pretrained knowledge and mitigate interference.

### C.4. Analysis of Rank Selection

In the paper, we use a single global rank shared across layers, which works well across reported settings. To study a more automated choice, we further evaluate a layer-wise Top-$p$ scheme, where for each layer, we adopt the smallest rank that retains $p\%$ of the total sum of singular values (spectral energy). As shown in Table 11, Top-$p$ performs slightly better than fixed rank.

## D. Analysis of Knowledge Conflict

TSV and ISO primarily focus on preserving the spectral structure of task vectors, whereas CSP explicitly targets cross-task interference by constraining updates to lie outside the pretrained core subspace. To understand the mechanism, we isolate the component removed by CSP, i.e., $Proj(T)$, the projection of the task update $T$ inside the pretrained core subspace.

As shown in Table 12, $Proj(T)$ can help some tasks but harm others, suggesting that it contains entangled, cross-task-conflicting changes rather than universally beneficial shared knowledge. CSP effectively removes this conflicting component,

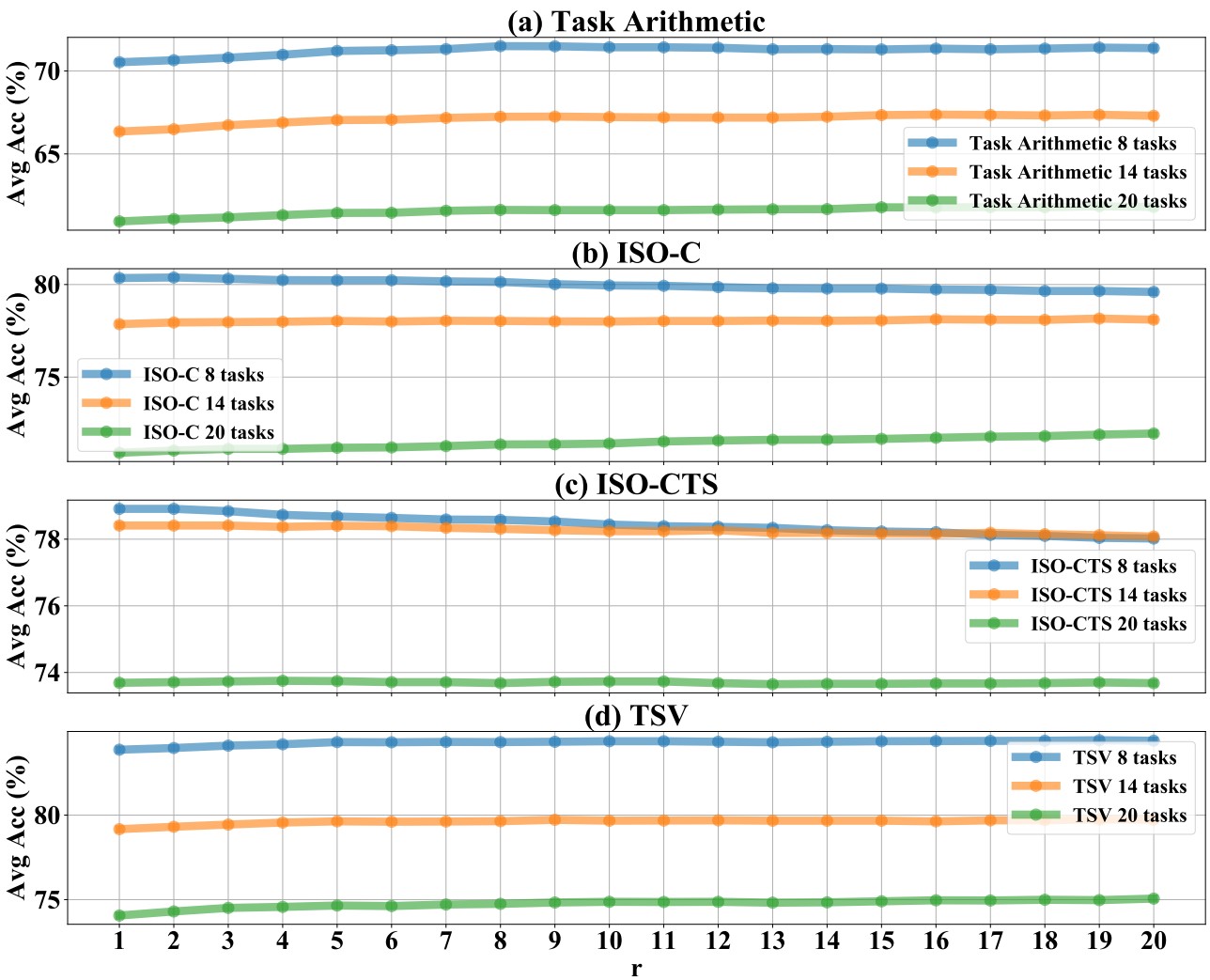

*Figure 8.* Average accuracy across different numbers of tasks with various $r$.

leading to more robust merging performance.

## E. Analysis of Projection Effect

CSP removes specific components of task vectors through projection. The subspace is not arbitrary: it is always defined by the top $r$ singular directions of the pretrained model. To better understand the effect of this choice, we conduct controlled comparisons using **Bottom-r** subspaces (spanned by the smallest singular directions) and 5 random $r$-dimensional subspaces.

As shown in Table 13, both Bottom-$r$ and random subspaces provide limited or inconsistent gains, whereas projecting onto the dominant top-$r$ directions consistently improves performance. This indicates that the observed improvement is not simply due to projection, but rather to constraining updates outside the dominant pretrained subspace, which is critical for mitigating cross-task interference.

## F. Analysis of Generalization Ability

To assess the generalization of CSP, we evaluate performance on unseen tasks using ViT-B/32. Table 14 reports accuracy on both seen and unseen tasks under different merging scenarios. The results indicate that CSP consistently improves performance, not only on tasks included in the merging process but also on previously unseen tasks, demonstrating that

*Table 12.* Effect of the component removed by CSP on individual tasks. Values indicate absolute accuracy (%).

|  | SUN397 | Cars | RESISC45 | EuroSAT | SVHN | GTSRB | MNIST | DTD |
|---|---|---|---|---|---|---|---|---|
| $W_{pre}$ | 62.3 | 59.7 | 60.7 | 45.5 | 31.4 | 32.6 | 48.5 | 43.8 |
| $W_{pre} + Proj(\tau_{ISOCTS})$ | -0.8 | -2.2 | -1.4 | +7.7 | +6.8 | -3.3 | +6.0 | -4.3 |
| $W_{pre} + Proj(\tau_{TSV})$ | -0.5 | -0.8 | -0.5 | +6.5 | +7.5 | -0.6 | +7.3 | -3.6 |

*Table 13.* Effect of different subspace choices on model merging performance (8 vision tasks, accuracy improvements in %). Values for Random indicate mean $\pm$ std over 5 trials.

| Method (Backbone) | Bottom-$r$ | Random | Top-$r$ |
|---|---|---|---|
| ISO-CTS (ViT-B/32) | +0.1 | -0.9 $\pm$ 1.10 | +0.4 |
| ISO-CTS (ViT-L/14) | +0.0 | -1.0 $\pm$ 0.57 | +0.4 |
| TSV (ViT-B/32) | +0.0 | -0.8 $\pm$ 0.50 | +1.0 |
| TSV (ViT-L/14) | -0.7 | -1.1 $\pm$ 0.49 | +0.8 |

preserving the pretrained core subspace enhances generalization across new downstream tasks.

## G. Analysis of BLIP Captioning

Hypothesis 4.1 is a conditional statement expected to hold when the first- and second-order conditions are satisfied. It may fail in two cases:

1. **Violation of the first-order condition:** the gradient component projected onto the core subspace is non-negligible, i.e.,

$$\nabla_W L_{\mathrm{mtl}}(W_{\mathrm{pre}}) \notin \Pi_r^{\perp}(W_{\mathrm{pre}}),$$

    where $\Pi_r^{\perp}(W_{\mathrm{pre}})$ denotes the orthogonal complement of the top-$r$ pretrained core subspace.

2. **Violation of the second-order condition:** there exists a direction $\Delta W \in \Pi_r(W_{\mathrm{pre}})$ such that

$$\langle \Delta W, \nabla^2 L_{\mathrm{mtl}}(W_{\mathrm{pre}})[\Delta W] \rangle < 0,$$

    indicating local concavity along the core subspace.

The BLIP captioning results can be viewed as a boundary case of type (i). For each layer $l$, we project the task gradient $G^l$ onto the pretrained core subspace and compute the ratio

$$r^l = \frac{\|\mathrm{Proj}(G^l)\|_F^2}{\|G^l\|_F^2}.$$

Table 15 reports the average ratio across layers in the vision encoder, text encoder, text decoder, and all modules.

Compared with VQA tasks, captioning tasks consistently exhibit a larger ratio of gradients within the core subspace, indicating that **the first-order condition is less well satisfied** for captioning. These ratios remain relatively small overall, suggesting that BLIP captioning lies near the boundary of Hypothesis 4.1. CSP introduces a trade-off on captioning metrics while substantially protecting VQA performance. This phenomenon is not specific to BLIP: any downstream task whose updates overlap significantly with the pretrained core subspace may show similar behavior.

Notably, the text decoder shows the largest gap, consistent with captioning requiring richer language generation than VQA.

## H. Theoretical Analysis of $U_r$ Importance in Ablation Study

The ablation study indicates that the left singular vectors $U_r$ are more important than the right singular vectors $V_r$ for preserving task performance. This observation can be theoretically explained by the alignment between task vectors and the pretrained subspaces.

*Table 14.* Effect of CSP on seen and unseen tasks (20 vision tasks, accuracy %). The first 8/14 tasks are considered seen, the remaining tasks are unseen.

| Method | Seen (first 8 tasks) | Unseen (last 12 tasks) | Seen (first 14 tasks) | Unseen (last 6 tasks) |
|---|---|---|---|---|
| ISO-CTS | 78.84 | 58.75 | 78.12 | 51.93 |
| ISO-CTS w/ ours | 78.91 | 58.90 | 78.41 | 52.20 |
| TSV | 83.08 | 55.04 | 78.38 | 50.68 |
| TSV w/ ours | 84.43 | 55.50 | 79.74 | 51.41 |

*Table 15.* Projection ratios of task gradients onto the pretrained core subspace for BLIP captioning and VQA tasks. Larger ratios indicate greater overlap with the core subspace and potential violation of the first-order condition.

| Module | COCO Caption | Flickr30k Caption | TextCaps | OKVQA | TextVQA | ScienceQA |
|---|---|---|---|---|---|---|
| Vision Encoder | 1.9% | 1.6% | 1.4% | 1.1% | 0.9% | 1.1% |
| Text Encoder | 2.7% | 3.1% | 3.6% | 1.3% | 1.4% | 1.4% |
| Text Decoder | 6.4% | 5.9% | 4.9% | 2.6% | 1.4% | 2.7% |
| All Modules | 4.0% | 3.6% | 3.6% | 1.8% | 1.3% | 1.9% |

*Table 16.* Projection of task vectors $T$ onto the left ($U_r$) and right ($V_r$) singular subspaces across datasets and backbones. Values are percentages of Frobenius norm squared.

| | Backbone | $\tau_{\text{SUN397}}$ | $\tau_{\text{Cars}}$ | $\tau_{\text{RESISC45}}$ | $\tau_{\text{EuroSAT}}$ | $\tau_{\text{SVHN}}$ | $\tau_{\text{GTSRB}}$ | $\tau_{\text{MNIST}}$ | $\tau_{\text{DTD}}$ | $\tau_{\text{SUM}}$ |
|---|---|---|---|---|---|---|---|---|---|---|
| $r_U$ | ViT-B/32 | 1.7 | 1.7 | 1.8 | 1.7 | 1.7 | 1.8 | 1.8 | 1.8 | 2.1 |
| $r_V$ | ViT-B/32 | 1.0 | 1.0 | 1.1 | 1.2 | 1.1 | 1.0 | 1.0 | 1.1 | 1.3 |
| $r_U$ | ViT-L/14 | 2.1 | 2.1 | 2.0 | 2.0 | 2.1 | 2.1 | 2.1 | 2.1 | 2.3 |
| $r_V$ | ViT-L/14 | 1.1 | 1.2 | 1.1 | 1.1 | 1.1 | 1.0 | 1.1 | 1.1 | 1.3 |

Before providing the theoretical reasoning, we first measure the projection energy of the task vector $T$ onto the two subspaces:

$$r_U = \frac{\|U_r U_r^\top T\|_F^2}{\|T\|_F^2}, \quad r_V = \frac{\|T V_r V_r^\top\|_F^2}{\|T\|_F^2}.$$

As shown in Table 16, $r_U$ is consistently larger than $r_V$ across datasets and backbones, indicating that task vectors are more strongly aligned with $U_r$ than with $V_r$.

**Theoretical explanation.** Consider a linear layer $y = xW$, where the gradient of the loss $L$ with respect to $W$ is

$$\nabla_W L = x^\top (\nabla_y L).$$

The task vector $T$ can be interpreted as the accumulation of gradients over fine-tuning steps. Therefore, its column space is approximately controlled by the span of input features:

$$\text{col-space}(T) \subseteq \text{span}\{x^{(1)\top}, \ldots, x^{(t)\top}\}.$$

Under the standard assumption in transfer learning that downstream features largely align with the dominant pretrained subspace, there exists $a_t$ such that

$$x^{(t)\top} = U a_t + \epsilon_t, \quad \|\epsilon_t\| \ll \|x^{(t)\top}\|,$$

which is consistent with the widely observed feature-reuse phenomenon (Lippl & Lindsey, 2024).

Substituting into $T$ yields

$$T = \sum_t x^{(t)\top} \nabla_y L^{(t)} = U \sum_t a_t \nabla_y L^{(t)} + \mathcal{O}(\epsilon) \approx UA,$$

showing that task vectors are primarily contained in the span of $U$. In contrast, no similar alignment is expected for the right singular vectors $V$, because output-side backpropagated signals depend more directly on task-specific labels and objectives. This induces an asymmetry: task vectors are naturally more aligned with $U_r$ than $V_r$.

Empirically, preserving both $U_r$ and $V_r$ still yields the best performance. Nevertheless, this asymmetry helps explain why the ablation study observed a stronger effect from $U_r$ than from $V_r$.

