# OpenReview forum: "Revisiting the Role of Pretrained Weights in Model Merging: On Near-Optimality within the Core Subspace"
_ICML.cc/2026/Conference — ICML 2026 regular_

### Official Review · Reviewer_rPcE · 2026-03-01

**Soundness:** 2
**Presentation:** 2
**Significance:** 3
**Originality:** 3
**Overall Recommendation:** 4
**Confidence:** 3

**Summary:**

This paper revisits the role of pretrained weights in the context of model merging. The authors propose that a pretrained model's parameters are near-optimal within a "core subspace" defined by its dominant singular directions. They provide empirical evidence showing that the multi-task loss gradient is negligible and the curvature is predominantly non-negative within this subspace, suggesting approximate first- and second-order stationarity. Based on this insight, they introduce Core Subspace Preservation that projects the merged task vector onto the orthogonal complement of this core subspace before adding it to the pretrained weights. Experiments on vision and vision-language benchmarks demonstrate that applying it consistently improves performance.

**Compliance With Llm Reviewing Policy:**

Affirmed.

**Final Justification:**

I appreciate the additional experimental results.

However, I still remain cautious about the theoretical analysis of $U_k$ that does not align with the results in the original paper.

**Key Questions For Authors:**

1. Can Hypothesis 4.1 be formalized and proved under specific, reasonable conditions? What would those conditions be?
Is there a theoretical justification for choosing the core subspace rank k, perhaps based on the spectrum of Wpre or anything else, to make it an adaptive one?

2. The ablation shows that the left singular vectors are most important. Does this imply the "core" property is more about stabilizing the model's output features? Could the method be simplified to only preserve Uk?

3. The gains are larger for simpler merging methods. Does this imply solving a specific type of interference? Can you characterize the conflicts it resolves versus those resolved by, say, TSV or ISO?

**Strengths And Weaknesses:**

Strength

1. The paper provides a principled, optimization-based lens to analyze pretrained weights beyond treating the pretrained model as a static anchor and offers a novel explanatory framework for its generalization properties.

2. The method is explicitly designed as a post-processing module applicable to any merged task vector.

3. The paper conduct experients on several models, tasks and backbones, notably multimodal tasks in Table 4.

4. The paper includes insightful ablations and sensitivity analyses.

Weakness
1. Compare with more recent subspace based method: some key method are missing[1,2].

2. The claim of "near-optimality" is conditioned on a manually selected subspace dimension. The connection between the approximate stationarity within the subspace and the final task performance gain, while intuitive, is not formally established. The improvement could be due to other effects of the projection.

3. The core subspace is defined purely via the SVD of Wpre, agnostic to the downstream tasks T. The hypothesis assumes the pretrained model has "sufficiently trained on relevant datasets" (Hypothesis 4.1). If the pretraining and downstream tasks are misaligned, the principal directions of Wpre may not be "core" for generalization. The failure on BLIP captioning hints at this limitation but is not analyzed from this perspective.

4. This paper claims generalization ability beyond pre-trained model, but the experiments still limit to traidtional setting. If possible, I suggest the author to try a generalization setting to train and merge on some tasks, and then validate them on some unseen tasks.

5. The improvement over strong baselines like ISO-CTS and TSV on ViT models, while consistent, is often marginal, questioning its practial application. Additionally, the paper does not provide a statistical significance test or error bars, making it harder to assess the substantiality of these improvements, especially relative to the variance of the tasks.

6. Selection of  hyper-parameter k varies with the architecture, number of tasks, which may weaken the statement of plug-and-play attribution.

[1] Model merging with SVD to tie the Knots. ICLR 2025.

[2] RobustMerge: Parameter-Efficient Model Merging for MLLMs with Direction Robustness. NeurIPS 2025.

---

> ### Author Rebuttal · Authors · 2026-03-31
>
> ## **Response to rPcE**
>
> **Q4.1. Compare related methods (KnOTS [1] and RobustMerge [2]).**
>
> Thanks for the suggestion. KnOTS and RobustMerge primarily focus on **merging task-specific knowledge in LoRA adapters**, while CS Preservation protects the core subspace to **reduce cross-task conflict**.
>
> As a plug-and-play module, our approach is complementary to these methods. Empirically, in the LoRA-based setting of KnOTS [1], CS Preservation increases the average accuracy of KnOTS from 54.51 to 56.38 when merging seven ViT-B/32 models, and also improves RobustMerge from 55.05 to 56.97 (see Q1.5 for task-specific results).
>
> **Q4.2. The performance gain may come from projection.**
>
> The subspace is not arbitrary: it is always spanned by the **top singular directions of $W_{pre}$**. We conduct controlled comparisons using Bottom-K subspaces (smallest-singular directions) and 5 random K-dimensional subspaces.
>
> As shown, both have limited gains, suggesting that the improvement is **not** due to arbitrary projection, but to the dominant pretrained subspace.
>
> |  | Bottom-K | Random | Top-K |
> | --- | --- | --- | --- |
> | ISO-CTS (ViT-B/32) | +0.1 | -0.9±1.10 | +0.4 |
> | ISO-CTS (ViT-L/14) | +0.0 | -1.0±0.57 | +0.4 |
> | TSV (ViT-B/32) | +0.0 | -0.8±0.50 | +1.0 |
> | TSV (ViT-L/14) | -0.7 | -1.1±0.49 | +0.8 |
>
> **Q4.3. Can Hypothesis 4.1 be formalized under reasonable conditions? If pretrained and downstream tasks are misaligned, how would CS Preservation behave?**
>
> We would like to clarify that Sec. 4.1 **characterizes** Hypothesis 4.1 through two conditions: (i) the gradient projected onto the core subspace is negligible, and (ii) the curvature within the core subspace is predominantly non-negative.
>
> Sec. 4.2 provides empirical evidence that these conditions hold in common settings. However, proving that they hold for general deep nonlinear networks remains challenging and is beyond the scope of this paper.
>
> These conditions are more likely to hold when (i) pretraining captures transferable structure, (ii) downstream tasks are sufficiently aligned with the pretraining distribution so that the multi-task optimum remains close to $W_{pre}$ along the dominant pretrained directions, and (iii) task-specific updates are concentrated in the orthogonal complement of this dominant subspace.
>
> The weaker results on BLIP captioning are consistent with the boundary of the hypothesis: when pretraining and downstream tasks are strongly misaligned, the dominant singular directions of $W_{pre}$ need not correspond to task-relevant “core” directions, and the benefit of CS Preservation becomes limited.
>
> **Q4.4. Evaluate the generalization on unseen tasks.**
>
> Following the suggestion, we evaluate unseen-task generalization on ViT-B/32. The results below show that CS Preservation improves performance in both seen and unseen tasks.
>
> |  | Seen (first 8 tasks) | Unseen (last 12 tasks) | Seen (first 14 tasks) | Unseen (last 6 tasks) |
> | --- | --- | --- | --- | --- |
> | ISO-CTS | 78.84 | 58.75 | 78.12 | 51.93 |
> | ISO-CTS w/ ours | **78.91** | **58.90** | **78.41** | **52.20** |
> | TSV | 83.08 | 55.04 | 78.38 | 50.68 |
> | TSV w/ ours | **84.43** | **55.50** | **79.74** | **51.41** |
>
> **Q4.5. Statistical significance tests are needed.**
>
> We conduct a controlled study. On ViT-B/32 with 8 tasks, we finetune new checkpoints 5 times, each with a random seed for 3,000 training steps. After applying CS Preservation, ISO-CTS improves from 78.64±0.2 to 79.29±0.4 (p≈0.03), and TSV improves from 80.48±0.3 to 81.85±0.4 (p≈0.0005). The p-values are computed using a two-sided paired t-test across seeds. Both gains are statistically significant.
>
> **Q4.6. The choice of k varies across settings. How to choose in practice?**
>
> In practice, k does not require an unconstrained search: we find that performance is stable within a relatively small range, and a coarse search over a logarithmic grid (e.g., {1, 2, 5, 10, 20}) is sufficient. Empirically, stronger task interference consistently favors a larger k.
>
> An adaptive strategy for selecting k may further improve performance, as evidenced by the results of the top-p-style criterion experiments in Q2.1.
>
> **Q4.7. Why is $U_k$ more important in the ablation study? Could it preserve only $U_k$?**
>
> A possible reason is that stabilizing the dominant feature-side structure is particularly important. Nevertheless, Table 5 shows that preserving both $U_k$ and $V_k$ is consistently better than preserving only one side, with essentially no extra cost.
>
> **Q4.8. Does CS Preservation resolve a specific conflict that TSV/ISO does not?**
>
> TSV/ISO aims to retain **task-specific** knowledge, while CS Preservation aims to reduce **cross-task** conflict. As shown in the experiments in **Q2.4**, CS Preservation removes components of task vectors that contain entangled, cross-task-conflicting changes rather than uniformly beneficial shared knowledge. CS Preservation removes precisely this conflicting component during merging.

---

> > ### Author Rebuttal · Reviewer_rPcE · 2026-04-03
> >
> > Thanks for the reply. Most of my concerns are solved. Remember to add the additional results in the revised manuscripts.
> >
> > My left questions would be:
> >
> > I'm not sure if the hypothesis in Q4.3 is too strong. Without knowing this, we are unable to know when the failure, as in caption would occur. Is it a common phenomenon or an isolated incident? A theoretical derivation rather than empirical results would be better.
> >
> > I'd be more satisfied if Q4.7 is analyzed from a theoretical perspective rather than an empirical aspect, since the empirical results are obvious to see.

---

> > > ### Author Response · Authors · 2026-04-05
> > >
> > > ## Response to rPcE
> > >
> > > Thanks for the follow-up. We are pleased that most concerns have been addressed. We will incorporate all the additional results into the revised manuscript.
> > >
> > > Below, we respond to the remaining questions.
> > >
> > > **Q: Is Hypothesis 4.1 too strong? When will it fail like the BLIP caption tasks? Is the BLIP caption task a common phenomenon or an isolated incident?**
> > >
> > > A: **Hypothesis 4.1 can be view as a conditional statement expected to hold when the first- and second-order conditions in Sec. 4.1 are satisfied.** It may fail when
> > >
> > > (i) the first-order condition is violated, i.e., the gradient component projected onto the core subspace is non-negligible,
> > > $\nabla_W L_{mtl}(W_{pre}) \notin \Pi^\perp_k(W_{pre})$, or
> > >
> > > (ii) the second-order condition is violated, i.e., there exists a direction $\Delta W \in \Pi_k(W_{pre})$ such that
> > >
> > > $\langle \Delta W,\nabla^2 L_{mtl}(W_{pre})[\Delta W]\rangle < 0.$
> > >
> > > **The BLIP captioning results can be viewed as a boundary case of type (i).** Specifically, for each layer $l$, we project the task gradient $G^l$ onto the pretrained core subspace and compute the ratio $||Proj(G^l)||^2_F / ||G^l||^2_F$. The table below reports the average ratio across layers in the vision encoder, text encoder, text decoder, and all modules.
> > >
> > > Compared with VQA tasks, captioning tasks consistently exhibit a larger ratio. In other words, **the first-order condition is less well satisfied for captioning**. These ratios remain relatively small overall, suggesting that BLIP captioning lies near the boundary of the hypothesis. **This is not specific to BLIP captioning: similar behavior may arise for any downstream task whose task-relevant updates have substantial overlap with the pretrained core subspace.**
> > >
> > > ||COCO Caption|Flickr30k Caption|TextCaps|OKVQA|TextVQA|ScienceQA|
> > > |-|-|-|-|-|-|-|
> > > |Vision Encoder|1.9%|1.6%|1.4%|1.1%|0.9%|1.1%|
> > > |Text Encoder|2.7%|3.1%|3.6%|1.3%|1.4%|1.4%|
> > > |Text Decoder|6.4%|5.9%|4.9%|2.6%|1.4%|2.7%|
> > > |All modules|4.0%|3.6%|3.6%|1.8%|1.3%|1.9%|
> > >
> > > Notably, the text decoder shows the largest gap. This is consistent with captioning requiring richer language generation than VQA.
> > >
> > > **Q: From a theoretical perspective, why is** $U_k$ **more important in the ablation study?**
> > >
> > > A: A theoretical explanation is that the task vector $T$ is typically more aligned with the $U_k$ directions than with the corresponding $V_k$ directions.  Before providing theoretical reasons, we measure the projection energy of $T$ onto the two subspaces:
> > >
> > > $r_U = \frac{||U_k U_k^\top T||^2_F}{||T||^2_F}, \quad r_V = \frac{||T V_k V_k^\top||^2_F}{||T||^2_F}.$
> > >
> > > The results below consistently show $r_U >r_V$ across datasets and backbones, indicating stronger alignment with $U_k$.
> > >
> > > ||Backbone|$T_{SUN397}$|$T_{Cars}$|$T_{RESISC45}$|$T_{EuroSAT}$|$T_{SVHN}$|$T_{GTSRB}$|$T_{MNIST}$|$T_{DTD}$|$T_{SUM}$|
> > > |-|-|-|-|-|-|-|-|-|-|-|
> > > |$r_U$|ViT-B/32|1.7%|1.7%|1.8%|1.7%|1.7%|1.8%|1.8%|1.8%|2.1%|
> > > |$r_V$|ViT-B/32|1.0%|1.0%|1.1%|1.2%|1.1%|1.0%|1.0%|1.1%|1.3%|
> > > |$r_U$|ViT-L/14|2.1%|2.1%|2.0%|2.0%|2.1%|2.1%|2.1%|2.1%|2.3%|
> > > |$r_V$|ViT-L/14|1.1%|1.2%|1.1%|1.1%|1.1%|1.0%|1.1%|1.1%|1.3%|
> > >
> > > ***A theoretical explanation is as follows.***  Consider a linear layer $y = xW$. The gradient of loss $L$ with respect to $W$ is $\nabla_W L =  x^\top(\nabla_y L)$ . The task vector $T$ can be viewed as the accumulation of gradients throughout fine-tuning. Therefore, its column space is controlled by the span of the input features
> > >
> > > $\mathrm{col\text{-}space}(T) \subseteq \mathrm{span}${$x^{(1)\top},\dots,x^{(t)\top}$}.
> > >
> > > We further adopt a standard assumption in transfer learning: downstream features are largely aligned with the dominant pretrained subspace, i.e., there exists a vector $a_t$ such that
> > >
> > > ${x^{(t)}}^\top =  U a_t  + \epsilon_t,\quad ||\epsilon_t|| \ll ||{x^{(t)}}^\top||.$
> > >
> > > This is consistent with the widely observed feature-reuse phenomenon, where pretrained representations remain largely stable and downstream adaptation mainly recombines existing features [1].
> > >
> > > Substituting into $T$, we obtain:
> > >
> > > $T =\sum_t {x^{(t)}}^\top \nabla_y L^{(t)} = U \sum_t a_t \nabla_y L^{(t)}   + \mathcal{O}(\epsilon),$
> > >
> > > which simplifies to
> > >
> > > $T \approx  U A.$
> > >
> > > By contrast, we do not in general expect a similar alignment on the output side $V$, because ***the output-side backpropagated signals depend more directly on the downstream objective and label space, which are typically more task-specific than the reused input features***. This induces an asymmetry: the task vector tends to be more tightly aligned with the $U_k$ subspace than with $V_k$.
> > >
> > > We thank the reviewer for highlighting this point. We find this asymmetry both interesting and worthy of more rigorous future investigation. Empirically, however, preserving both $U_k$ and $V_k$ still yields the best overall performance, so we will keep our method unchanged in the revised manuscript.
> > >
> > > [1] Inductive biases of multi-task learning and finetuning: multiple regimes of feature reuse. NeurIPS, 2024.

---

### Official Review · Reviewer_imAC · 2026-03-10

**Soundness:** 3
**Presentation:** 3
**Significance:** 3
**Originality:** 3
**Overall Recommendation:** 5
**Confidence:** 4

**Summary:**

The paper explores the role of pre-trained weights in model merging, particularly the role of the core subspace, i.e., the subspace spanned by the leading eigenvectors. The authors first propose a mathematical formulation stating that, under certain assumptions, the pre-trained weights constitute a local minimizer of the multi-task loss. Afterwards, they verify empirically that these assumptions (e.g., the non-negativity of the Hessian) hold, thereby validating their framework. On top of that, they propose a simple approach that preserves the information encoded in the core subspace. Experiments show that the proposed approach, which can be conceived as an additional post-processing step on top of existing methods, leads to consistent and meaningful gains in model merging.

**Compliance With Llm Reviewing Policy:**

Affirmed.

**Final Justification:**

All of my concerns have been satisfactorily addressed, and I appreciate the clarifications and additional details provided.

**Key Questions For Authors:**

See above.

**Limitations:**

No, I did not find an adequate discussion of the limitations of the proposed formulation. The authors should address this issue more explicitly.

**Strengths And Weaknesses:**

In general terms, the paper represents a valuable theoretical contribution in a field (model merging) where empirical results are often not adequately explained or supported by rigorous mathematical analysis. The approach followed by the authors is sound: they formulate a set of assumptions to make the problem more tractable and then show that these assumptions hold to a certain extent. Moreover, they leverage these findings to derive a practical method for improving existing approaches. Hence, the paper is well structured. Additionally, addressing model merging from the perspective of the pre-trained model is noteworthy and relatively underexplored, as most existing works focus instead on post-hoc merging strategies.

Considering the weaknesses, my concerns are mainly related to the writing. Some passages could be significantly refined. Below, I provide a list of the sections that I found difficult to follow or that could be improved.

0. As a major point, the link between Sec. 4 and Sec. 5 is unclear and insufficiently emphasized. I do not fully understand the connection between the near-optimality of W_pre and the claim that task-specific knowledge should be injected primarily into the orthogonal complement. While this implication may be immediate to the authors, it is not obvious to the reader.
1. Lines 111–112 — Eq. 5. The authors introduce a formal description of the core subspace as the set of matrices W satisfying (I - UU^T) W (I - VV^T) = 0, where U and V are obtained from the pre-training weights. The text would benefit from an explanation or interpretation of this space to help the reader better visualize the definition, especially given its importance in the subsequent discussion.
2. In Sec. 4.1, the authors derive two necessary conditions to conclude that the pre-trained weights constitute a local minimizer. It would be helpful to explicitly highlight these two resulting conditions, for instance in a boxed summary at the end of the section. This would allow the reader to more easily revisit the derivations and better interpret the subsequent experiments.
3. More generally, I found some confusion in understanding whether the adaptation is intended to rely on directions within the core subspace or outside it. I encourage the authors to clarify this point across the whole paper to improve readability.
4. When assessing the loss landscape, the authors restrict the sampling of perturbations to the core subspace. Why was this choice made, instead of exploring a uniform neighborhood around the pre-trained weights? Moreover, the authors should provide more details on how the two plots in Figure 2(b) and (c) are generated (e.g., what the x- and y-axes represent).
5. It is not clear whether the reported accuracy values are absolute or normalized. The authors should clarify this point and ideally provide both, as is common practice in the field.
6. The authors should discuss the relation to [A], which also investigates the near-optimality of pre-trained weights from a second-order perspective.

[A] Porrello et al. (2024), “A Second-Order Perspective on Model Compositionality and Incremental Learning” (ICLR 2025)

Minor.

The authors use T_k to denote the task vector and tau_k to denote the task itself. In my experience, the opposite convention is more commonly adopted.

---

> ### Author Rebuttal · Authors · 2026-03-31
>
> ## **Response to imAC**
>
> **Q3.1. The link between Sec. 4 and Sec. 5 should be described more explicitly.**
>
> Thanks for the suggestion. In Sec. 4, we observe a key implication: within the pretrained core subspace, $W_{pre}$ behaves as a local minimizer of the multi-task objective. The reasons are twofold:
>
> - Gradients are negligible;
> - Curvature is largely non-negative.
>
> Consequently, **updates along this subspace are unlikely to yield gains and instead risk disrupting shared pretrained knowledge**.
>
> This directly **motivates** our design in Sec. 5: preserve the core subspace and inject task-specific updates into its orthogonal complement. We will revise Secs. 4–5 to make this causal link explicit.
>
> **Q3.2. This paper should include an explanation for Eq. (5).**
>
> We will add an intuitive explanation after Eq. (5).
>
> For any matrix $W$, $U_k$ and $V_k$ induce a four-block decomposition:
>
> $W = U_kU_k^\top W V_kV_k^\top + (I - U_k U_k^\top) W V_kV_k^\top + U_kU_k^\top W (I - V_k V_k^\top) + (I - U_k U_k^\top) W (I - V_k V_k^\top)$
>
> Eq. (5) defines the core subspace as
>
> $\Pi_k(W_{pre}) \triangleq ${$ W | (I - U_k U_k^\top)\  W\ (I - V_k V_k^\top)=0 $}
>
> This means that the last block in the above decomposition is zero. Equivalently, $W$ may vary only along directions aligned with the principal singular structure of $W_{pre}$.
>
> Intuitively, updates within this subspace affect the principal components of $W_{pre}$ . Therefore, this is precisely the part we aim to preserve during merging.
>
> **Q3.3. The two necessary conditions in Sec. 4.1 should be summarized more explicitly.**
>
> We agree. In the revision, we will add a **boxed** summary highlighting:
>
>  (1) First-order condition: the gradient at $W_{pre}$ has negligible projection onto the core subspace;
>
>  (2) Second-order condition: curvature along directions in the core subspace is predominantly non-negative.
>
> **Q3.4. It is unclear whether adaptation should be inside or outside the core subspace.**
>
> We will make this consistent throughout:
>
> - The pretrained core subspace should be **preserved.**
> - Task-specific adaptation should be injected **outside** the core subspace (i.e., in the orthogonal complement).
>
> This is the central design principle of our method and will be stated explicitly wherever relevant.
>
> **Q3.5. Why restrict loss-landscape perturbations to the core subspace? How are Figures 2(b) and (c) generated?**
>
> The goal of Sec. 4 is to test near-optimality **within the core subspace**. Sampling perturbations in the full parameter space would mix core and non-core directions, obscuring this property.
>
> For the generation of Figs. 2(b)(c), we construct a 2D plane spanned by three anchor points:
>
> - the pretrained model $W_{pre}$ (set to (0, 0.5)),
> - the multi-task trained model $W_{mtl}$ (set to (1, 0.5)),
> - and a core-subspace perturbation $W_{pre}+\alpha \sum_i u_i v_i^\top$ or $W_{pre}+\alpha u_{max} v_{max}^\top$ (set to (0, 1)).
>
> We then densely interpolate across this plane. Each interpolated point defines a model whose multi-task loss is evaluated, producing the loss landscape and contour plots. Accordingly, **the x- and y-axes are dimensionless and indicate only the interpolation coefficients among these anchor points**, not absolute parameter magnitudes.
>
> **Q3.6. Are the reported accuracies absolute or normalized?**
>
> All reported results are **absolute** metrics. We will also provide normalized metrics in the revision for completeness.
>
> **Q3.7. How does your work relate to Porrello et al. (2024)?**
>
> Thank you for suggesting this interesting work. We will discuss Porrello et al. (2024) in the revision. Both their work and ours concern the near-optimality of pretrained weights from a second-order perspective. Porrello et al. (2024) study this phenomenon in the context of **continual learning**, whereas our work focuses on **model merging**.
>
> More importantly, Porrello et al. assume that pretrained weights are near-optimal and that the loss landscape is approximately convex in the **whole parameter space**. In contrast, our paper investigates this property more explicitly: we show that the pretrained model exhibits local optimality within **a core subspace** spanned by its dominant singular directions, and we **provide empirical evidence** from both first-order and second-order conditions to support this claim.
>
> **Q3.8. The notation uses $T_k$ for task vectors and $\tau_k$ for tasks, while the opposite convention is more common.**
>
> Thanks for the suggestion. We will use more standard task/task-vector notation consistently throughout the paper.
>
> **Q3.9. The paper does not discuss limitations explicitly enough.**
>
> We agree and will add a dedicated limitations discussion.

---

> > ### Author Rebuttal · Reviewer_imAC · 2026-04-02
> >
> > I thank the authors for their thorough and helpful rebuttal. All of my concerns have been satisfactorily addressed. I will raise my score accordingly.

---

> > > ### Author Response · Authors · 2026-04-02
> > >
> > > Thank you for your positive feedback and for acknowledging that your concerns have been addressed. We appreciate your time and consideration.

---

### Official Review · Reviewer_AdbL · 2026-03-12

**Soundness:** 3
**Presentation:** 3
**Significance:** 2
**Originality:** 2
**Overall Recommendation:** 4
**Confidence:** 4

**Summary:**

This paper investigates the role of pretrained weights in the model merging process, shifting the focus from simply resolving conflicts between task vectors to preserving the intrinsic generalization capabilities of the pretrained model. The authors hypothesize that pretrained weights are "near-optimal" within a **core subspace** defined by their dominant singular vectors. Based on these insights, they propose **Core Subspace Preservation**, a training-free, plug-and-play module that projects merged task vectors onto the orthogonal complement of the pretrained core subspace. Experiments across vision (ViT) and vision-language (BLIP) models demonstrate that CS Preservation consistently enhances the performance of some existing state-of-the-art merging methods.

**Compliance With Llm Reviewing Policy:**

Affirmed.

**Key Questions For Authors:**

(1) In your experiments, did you find that a single global $k$ worked well for all layers, or did you vary $k$ based on the layer's depth or type?

(2) Since LoRA often targets low-rank updates, how does the "core subspace" of the pre-trained weights overlap with the subspaces typically learned during LoRA fine-tuning?

**Limitations:**

No, the authors have **not** explicitly discussed the limitations of their work in the main body or the supplemental materials. The followings are some suggestions.

(1) Discuss the sensitivity of the performance to the rank $k$. Mention that finding the optimal $k$ currently requires empirical tuning and may vary significantly across different model architectures or task distributions.

(2) Although described as "lightweight," performing Singular Value Decomposition (SVD) on every layer can become computationally expensive as model sizes scale to tens or hundreds of billions of parameters (e.g., large LLMs).

(3) Address whether the "near-optimality" of pretrained weights holds for tasks that are significantly "out-of-distribution" compared to the pre-training data, as the core subspace might not capture relevant features for such tasks.

**Strengths And Weaknesses:**

### **Strengths**

(1) While most merging research focuses on the properties of task vectors (fine-tuned minus pretrained), this work provides a principled explanation for the importance of the *pretrained* weights' spectral structure.

(2) The proposed method is a simple projection that requires no training or additional data. Its ability to be "plugged into" multiple existing frameworks makes it highly practical.

(3) The results in Tables 1 and 2 show that CS Preservation provides a performance boost across different backbones (ViT-B/32, ViT-L/14) and tasks (8 to 20 datasets).


### **Weaknesses**

(1) The effectiveness of the method depends on the choice of $k$ (the rank of the core subspace). While the authors discuss that larger models or more tasks favor larger $k$, the selection process remains somewhat empirical. A more automated or principled way to determine $k$ per layer would strengthen the work.

(2) While the authors describe the overhead as "minimal," performing SVD on all layers of extremely large models (e.g., 70B+ parameter LLMs) can be non-trivial. The current evaluation is limited to ViT and BLIP; extending this to modern LLMs would demonstrate broader scalability.

(3) Existing methods like TSV and ISO-Merging already leverage spectral properties. While the paper argues CS Preservation is complementary by focusing on $W_{pre}$ rather than $T^*$, the distinction in performance gains could be more explicitly dissected to show exactly what CS Preservation captures that TSV/ISO misses.

---

> ### Author Rebuttal · Authors · 2026-03-31
>
> ## **Response to AdbL**
>
> **Q2.1. A more automated way to determine $k$ would strengthen the work.**
>
> In the paper, we use a single global rank shared across layers, which works well across reported settings. To study a more automated choice, we further evaluate a layer-wise Top-P scheme, where for each layer, we adopt the smallest rank that retains P% of the total sum of singular values (spectral energy).
>
> Top-P performs slightly better than Top-K:
>
> | **Base Methods** | **Core Space** | **ViT-B/32** | **ViT-L/14** |
> | --- | --- | --- | --- |
> | ISO-CTS | Top-K | 84.7 | **93.4** |
> | ISO-CTS | Top-P | **84.8** | **93.4** |
> | TSV | Top-K | 85.1 | **92.5** |
> | TSV | Top-P | **85.2** | **92.5** |
>
> **Q2.2. Is the overhead still “minimal” for very large models like 70B+ LLMs?**
>
> CS Preservation requires a one-time SVD of each pretrained layer. In contrast, strong baselines such as ISO/TSV already involve repeated spectral operations. In our measurements (Table 8), CS Preservation (5s for ViT-B/32) is much cheaper than TSV/ISO (41s or 28s for ViT-B/32):
>
> | **Method** | **ISO-CTS** | **TSV** | **CS Preservation** |
> | --- | --- | --- | --- |
> | ViT-B/32 | 41 | 28 | **5** |
> | ViT-L/14 | 134 | 95 | **10** |
>
> Additionally, on Qwen2.5-72B, CS Preservation took **2.2 hours** on an RTX 3090 or **0.8 hours** on an H100. We agree that this overhead is not negligible at 70B+ scale, and will state this limitation explicitly; our point is that it is a one-time preprocessing cost and remains smaller than the repeated spectral operations used by ISO/TSV.
>
> **Q2.3. Evaluation on modern LLMs.**
>
> We evaluate CS Preservation on Qwen3-8B. The results show that CS Preservation is effective for LLM merging. Additional results on Qwen3-0.6B and Task Arithmetic are provided in Q1.5, showing similar trends.
>
> |  |  | **ArguAna** | **HotpotQA** | **BIOSSES** | **Banking77** |
> | --- | --- | --- | --- | --- | --- |
> |  |  | NDCG@10 | NDCG@10 | Pearson Correlation | ACC |
> | TSV | Qwen3-8B | 0.72 | 0.75 | 0.8644 | 84.72% |
> | TSV w/ ours | Qwen3-8B | **0.73** | **0.76** | **0.8665** | **84.83%** |
>
> **Q2.4. Does CS Preservation resolve a conflict that TSV/ISO does not?**
>
> TSV/ISO primarily focuses on preserving the **spectral structure of task vectors**, whereas CS Preservation targets **cross-task interference** by constraining updates to be outside the pretrained core subspace.
>
> We isolate the component **removed** by CS Preservation, i.e., $Proj(T)$ or the projection of $T$ inside the pretrained core subspace. As shown, $Proj(T)$ helps some tasks but hurts others, indicating that it contains entangled, cross-task-conflicting changes rather than uniformly beneficial shared knowledge. CS Preservation removes precisely this conflicting component.
>
> |  | **SUN397** | **Cars** | **RESISC45** | **EuroSAT** | **SVHN** | **GTSRB** | **MNIST** | **DTD** |
> | --- | --- | --- | --- | --- | --- | --- | --- | --- |
> | $W_{pre}$ | 62.3 | 59.7 | 60.7 | 45.5 | 31.4 | 32.6 | 48.5 | 43.8 |
> | $W_{pre} + Proj(T_{ISOCTS})$ | -0.8 | -2.2 | -1.4 | +7.7 | +6.8 | -3.3 | +6.0 | -4.3 |
> | $W_{pre} + Proj(T_{TSV})$ | -0.5 | -0.8 | -0.5 | +6.5 | +7.5 | -0.6 | +7.3 | -3.6 |
>
> **Q2.5. Applicability to LoRA?**
>
> LoRA imposes a low-rank structure on task-specific updates, but low rank does not imply orthogonality to the pretrained core subspace.
>
> We evaluate CS Preservation in LoRA-based merging of KnOTS [1]. We merge seven ViT-B/32 models, excluding SUN397 due to the reproducibility issue (#13) in the official repo. As shown (absolute accuracy, different from [1]), CS Preservation improves LoRA merging (see Q1.5 for task-specific results):
>
> |  | **Avg. Acc.** |
> | --- | --- |
> | Knots | 54.51 |
> | Knots w/ ours | **56.38** |
> | RobustMerge | 55.05 |
> | RobustMerge w/ ours | **56.97** |
>
> [1] Model merging with SVD to tie the knots. ICLR. 2025.
>
> **Q2.6.** **Does the "near-optimality" of** $W_{pre}$ **hold for OOD tasks?**
>
> We agree that OOD scenarios are challenging. The optimality of our method is conditioned on the pretrained model’s ability to capture transferable features relevant to the target tasks—a common assumption in model-merging research.
>
> Our BLIP results help clarify this boundary: CS Preservation provides clear gains on VQA tasks, where pretrained transfer is stronger. But it delivers limited gains on captioning tasks, where the pretrained BLIP model has weak zero-shot capability. This suggests that when pretraining and downstream tasks are strongly misaligned, the dominant singular directions of $W_{pre}$ need not correspond to task-relevant “core” directions, and the benefit of CS Preservation may diminish. We will state this limitation more explicitly in the revision.
>
> **Q2.7. Discuss limitations more explicitly?**
>
> We heartily agree. We will include a section on limitations, including rank selection, SVD cost at large scales, and reduced benefit under misaligned pretraining and downstream tasks.

---

> > ### Author Rebuttal · Reviewer_AdbL · 2026-04-01
> >
> > The authors have addressed my concerns. I will raise the score from 3 to 4.

---

> > > ### Author Response · Authors · 2026-04-01
> > >
> > > Thank you for your careful reading and for acknowledging that your concerns have been addressed. We appreciate your time and consideration.

---

### Official Review · Reviewer_fggS · 2026-03-12

**Soundness:** 3
**Presentation:** 3
**Significance:** 2
**Originality:** 2
**Overall Recommendation:** 5
**Confidence:** 4

**Summary:**

This paper investigates model merging from a subspace perspective, revealing that the core subspace of pretrained weights is essential for maintaining task generalization. The authors provide empirical and theoretical evidence that task-specific adaptations should be projected to the orthogonal complement of this subspace. Extensive experiments across multiple tasks demonstrate that the subspace-aware strategy consistently leads to improvement over existing weights merging methods.

**Compliance With Llm Reviewing Policy:**

Affirmed.

**Final Justification:**

My concerns have been addressed, and I’ll increase the score to 5.

**Key Questions For Authors:**

My major concern is about the effectiveness and robustness of the proposed method across a broader range of downstream tasks and backbones. For others, please refer to the weakness section.

**Limitations:**

Limitations are not discussed in the paper.

**Strengths And Weaknesses:**

Strengths
* The motivation of the paper is clear, and the findings offer significant theoretical value while being intuitively reasonable.
* The proposed method is both simple and effective; it is well-designed and consistently yields performance improvements

Weaknesses
* The notation k of the rank in line 110 should be changed to avoid ambiguity, as it is already used to denote the number of downstream tasks.
* I suggest including a brief description of the pre-trained backbones (e.g., CLIP or others) for each method in the settings section to ensure clarity.
* Comparing Table 1 and Table 2, the performance gains for larger-scale models appear limited. Could the authors provide a justification for this phenomenon?
* As shown in Figure 3, the optimal value for λ varies across different models, which suggests that the method is sensitive to this hyperparameter, potentially limiting its generalizability. Are there any strategies to address this sensitivity?

---

> ### Author Rebuttal · Authors · 2026-03-31
>
> ## **Response to fggS**
>
> **Q1.1. The notation $k$ is ambiguous, which denotes both the task index of tasks and the core-subspace rank.**
>
> Thank you for pointing this out. In the revision, we will reserve the subscript $k$ for task-related notation (e.g., $T_k$, $W_k$) and use a different symbol **$r$** for the core-subspace rank. We will update the corresponding definitions, equations, figures, and captions accordingly.
>
> **Q1.2. The paper should clarify the pretrained backbones used in the experiments.**
>
> Thanks for the suggestion. We will clarify this in Sec. 6.1.  The backbone details are as follows:
>
> - For vision tasks, we use CLIP vision encoders with ViT-B/32, ViT-B/16, and ViT-L/14 as the pretrained backbones. Task vectors are taken from the official repositories of Task Arithmetic (Tables 1–2) and FusionBench (Table 3).
> - For vision-language tasks, we use BLIP as the pretrained model and fine-tune all components (image encoder, text encoder, and text decoder) to obtain task vectors.
>
> **Q1.3. Why are the gains on larger-scale models in Table 2 smaller than those in Table 1?**
>
> We believe this is mainly because stronger backbones leave less room for improvement. In Table 2, strong baselines already yield competitive performance when merging ViT-L/14 models: ISO-CTS reaches 93.0, and TSV reaches 91.7, both close to the traditional MTL performance of 93.5. Under this regime, even modest performance gains are meaningful.
>
> This does not indicate reduced effectiveness on larger models. In the more challenging 14-task and 20-task settings in Table 3, CS Preservation still yields consistent gains on ViT-L/14. We will clarify this point in the revision.
>
> **Q1.4. Figure 3 suggests that the optimal $λ$ varies across models. Is the method sensitive to this hyperparameter?**
>
> We would like to clarify that **$\lambda$ is not introduced by our method**. It is a standard scaling coefficient inherited from baseline methods (e.g., Task Arithmetic, TSV).
>
> In fact, Figure 3 shows that after incorporating CS Preservation, the performance curves become more stable across a wide range of $\lambda$, indicating that CS Preservation improves robustness to the choice of $\lambda$.
>
> **Q1.5. How robust is the method across broader downstream tasks and backbones?**
>
> We agree that broader validation is important. We additionally evaluate CS Preservation on LLM merging with Qwen3-0.6B and Qwen3-8B. For each scale, we merge two official fine-tuned variants (Qwen3-Embedding and Qwen3-Reranker) and evaluate on two ranking tasks (ArguAna, HotpotQA) and two embedding tasks (BIOSSES, Banking77). The results below show mostly improvements across both scales, suggesting that CS Preservation remains effective for LLM merging.
>
> |  | **Backbone** | **ArguAna** | **HotpotQA** | **BIOSSES** | **Banking77** |
> | --- | --- | --- | --- | --- | --- |
> |  |  | NDCG@10 | NDCG@10 | Pearson Correlation | Accuracy |
> | Task Arithmetic | Qwen3-0.6B | 0.55 | 0.50 | 0.8324 | 74.67% |
> | Task Arithmetic w/ ours | Qwen3-0.6B | **0.57** | **0.56** | **0.8356** | 74.16% |
> | TSV | Qwen3-0.6B | 0.61 | 0.60 | 0.8235 | 75.51% |
> | TSV w/ ours | Qwen3-0.6B | **0.63** | 0.60 | **0.8266** | **76.03%** |
> | Task Arithmetic | Qwen3-8B | 0.63 | 0.64 | 0.8542 | 80.97% |
> | Task Arithmetic w/ ours | Qwen3-8B | 0.63 | **0.65** | **0.8625** | **81.43%** |
> | TSV | Qwen3-8B | 0.72 | 0.75 | 0.8644 | 84.72% |
> | TSV w/ ours | Qwen3-8B | **0.73** | **0.76** | **0.8665** | **84.83%** |
>
> In addition, we evaluate our method in the **LoRA-based merging setting** following KnOTS [1]. We merge seven ViT-B/32 models (excluding SUN397 due to reproducibility issues #13 reported in the official repository). As shown below, CS Preservation increases the average accuracy of KnOTS from 54.51 to 56.38, and also improves RobustMerge from 55.05 to 56.97. We report absolute accuracy, following our manuscript, rather than the normalized accuracy used in [1]. We will include these results in the revision.
>
> |  | **Cars** | **DTD** | **EuroSAT** | **GTSRB** | **MNIST** | **RESISC45** | **SVHN** | **Average** |
> | --- | --- | --- | --- | --- | --- | --- | --- | --- |
> | Knots[1] | 61.31 | 42.87 | 49.44 | 45.13 | 68.47 | 62.65 | 51.66 | 54.51 |
> | Knots w/ ours | **62.12** | **43.72** | **54.33** | 44.98 | **72.09** | **64.02** | **53.37** | **56.38** |
> | RobustMerge[2] | 61.93 | 42.38 | 52.34 | 44.79 | 69.22 | 61.49 | 53.23 | 55.05 |
> | RobustMerge w/ ours | **63.01** | **43.44** | **55.53** | **45.19** | **72.34** | **63.39** | **55.87** | **56.97** |
>
> References
>
> [1] Model merging with SVD to tie the knots//ICLR2025.
>
> [2] RobustMerge: Parameter-Efficient Model Merging for MLLMs with Direction Robustness// NeurIPS 2025.

---

> > ### Author Rebuttal · Reviewer_fggS · 2026-04-02
> >
> > Thanks author for the thorough response. My concerns have been addressed, and I’ll increase the score to 5.

---

> > > ### Author Response · Authors · 2026-04-02
> > >
> > > Thank you for your positive feedback and for confirming that your concerns have been fully addressed. We appreciate your time and consideration.

---

### Decision · Program_Chairs · 2026-04-30

**Decision:**

Accept (regular)

**Comment:**

The paper is clear and well supports its findings on the necessity of pre-trained weights for model merging.
While officially, no experiments were allowed during the rebuttal, many were made and other explanations (e.g. about related work) were made. I suggest incorporating those works and answering the questions reviewers made in the updated version of the paper.